# Efficient human-like antibody repertoire and hybridoma production in trans-chromosomic mice carrying megabase-sized human immunoglobulin loci

Hiroyuki Satofuka [1,9], Satoshi Abe [1,2,9], Takashi Moriwaki[1,3,9], Akane Okada[1], Kanako Kazuki[1], Hiroshi Tanaka[2], Kyotaro Yamazaki[4], Genki Hichiwa [1,3], Kayoko Morimoto[2], Haruka Takayama[2], Yuji Nakayama[5], Shinya Hatano [6], Yutaro Yada[6], Yasufumi Murakami[7], Yoshihiro Baba[6], Mitsuo Oshimura[2], Kazuma Tomizuka[8] & Yasuhiro Kazuki[1,3,4 ✉]

Trans-chromosomic (Tc) mice carrying mini-chromosomes with megabase-sized human immunoglobulin (Ig) loci have contributed to the development of fully human therapeutic monoclonal antibodies, but mitotic instability of human mini-chromosomes in mice may limit the efficiency of hybridoma production. Here, we establish human antibody-producing Tc mice (TC-mAb mice) that stably maintain a mouse-derived, engineered chromosome containing the entire human Ig heavy and kappa chain loci in a mouse Ig-knockout background. Comprehensive, high-throughput DNA sequencing shows that the human Ig repertoire, including variable gene usage, is well recapitulated in TC-mAb mice. Despite slightly altered B cell development and a delayed immune response, TC-mAb mice have more subsets of antigen-specific plasmablast and plasma cells than wild-type mice, leading to efficient hybridoma production. Our results thus suggest that TC-mAb mice offer a valuable platform for obtaining fully human therapeutic antibodies, and a useful model for elucidating the regulation of human Ig repertoire formation.

[1] Chromosome Engineering Research Center, Tottori University, 86 Nishi-cho, Yonago, Tottori 683-8503, Japan. [2] Trans Chromosomics Inc., 86 Nishi-cho, Yonago, Tottori 683-8503, Japan. [3] Division of Genome and Cellular Functions, Department of Molecular and Cellular Biology, School of Life Science, Faculty of Medicine, Tottori University, 86 Nishi-cho, Yonago, Tottori 683-8503, Japan. [4] Biomedical Science, Institute of Regenerative Medicine and Biofunction, Graduate School of Medical Science, Tottori University, 86 Nishi-cho, Yonago, Tottori 683-8503, Japan. [5] Division of Radioisotope Science, Research Initiative Center, Organization for Research Initiative and Promotion, Tottori University, Yonago 683-8503, Japan. [6] Division of Immunology and Genome Biology, Department of Molecular Genetics, Medical Institute of Bioregulation, Kyushu University, Higashi-ku, Fukuoka 812-8582, Japan. [7] Order-made Medical Research Inc., 5-4-19 Kashiwanoha, Kashiwa, Chiba 277-0882, Japan. [8] Laboratory of Bioengineering, Tokyo University of Pharmacy and Life Sciences, 1432-1 Horinouchi, Hachioji, Tokyo 192-0392, Japan. [9] These authors contributed equally: Hiroyuki Satofuka, Satoshi Abe, Takashi Moriwaki. ✉email: kazuki@tottori-u.ac.jp

In the past two decades, therapeutic antibodies (Abs) have emerged as a highly effective and fast-growing pharmaceutical option. Genetic engineering has been used to develop various methods to overcome the immunogenicity of rodent monoclonal antibodies (mAbs) in humans, which was a critical issue in early clinical trials. Transgenic animals designed to express the human Ab repertoire are widely recognized for their ability to generate fully human mAbs[1,2]. The first generation of humanized immunoglobulin (Ig) mice was developed in the 1990s by random integration of DNA segments containing partial human Ig heavy and light chain loci into the chromosomes of endogenous Ig-knockout mice[2,3]. These transgenic-knockout approaches revealed that diversification and selection in integrated human Ig loci are controlled by the animal's immune system and that they undergo natural processes of V(D)J rearrangement, somatic hypermutation (SHM) and class-switching. Furthermore, substantial efforts have been made to increase the number of V gene segments in mice, which is essential to produce a diverse repertoire of antigen-specific human antibodies and for proper development of the B cell lineage[2]. With such transgenic-knockout mice, antigen-specific fully human mAbs can be readily produced by well-established hybridoma technology[1].

This transgenic method, however, relies on random insertion of transgenes, making it difficult to include all regulatory elements[4], and the possibility that expression of the inserted human Ig loci will be affected by surrounding sequences exists. Recently, mice that produce a chimeric antibody comprising human variable and mouse constant regions have been engineered by a sophisticated method of replacing the genomic sequence of the mouse Ig variable region with that of a human Ig variable region[5–8]. In these mice, antigen-specific chimeric antibodies with human Ig variable regions are produced as efficiently as in wild-type (WT) mice, and by having a mouse-derived constant region, improved B cell development was achieved[5,7].

To introduce megabase-sized segments of DNA into mice, we have developed an alternative strategy utilizing a human chromosome as a vector for transgenesis[9]. Using this technology, two transmittable human chromosome fragments, one containing the Ig heavy chain locus (IGH, ≈1.5 Mb) and the other containing the kappa light chain locus (IGK, ≈2 Mb), were introduced into a mouse strain whose endogenous Igh and Igk loci were inactivated[10,11]. Hybridomas producing antigen-specific fully human antibodies were obtained from these trans-chromosomic (Tc) mice. Compared with other models, the double-Tc mice contained the largest fraction of human Ig loci at that time; however, some instability of human chromosome 2 (hChr.2)-derived human chromosome fragments containing IGK existed, contributing to lower hybridoma production efficiency, which was less than one-tenth of that observed in WT mice[12]. Additionally, human Ig repertoire formation that relies on introducing entire human Ig loci into mice remains to be evaluated in double-Tc mice. To solve this issue, a Tc mouse carrying hChr.14-derived fragment (hCF14) containing IGH was cross-bred with a YAC-transgenic mouse carrying ~50% of IGK segments, resulting in a new mouse strain exhibiting considerably improved hybridoma production[12]. However, subsequent studies revealed mosaicism of hCF14 in various tissues of Tc mice, indicating mitotic instability of the human centromere contained in hCF14[13,14]. We therefore constructed a mouse artificial chromosome (MAC) containing a mouse-derived centromere to improve the stability in Tc mice[9,15,16]. We demonstrated nearly perfect stability in all tissues of Tc mice, germline transmission to offspring and introduced exogenous gene expression[9]. Thus, the generation of MAC-based, human antibody-producing Tc mice has been anticipated.

In this study, we used a newly constructed artificial chromosome designated as IGHK-NAC to establish a fully human Ab-producing Tc mouse that efficiently produces antigen-specific Abs while recapitulating the human Ig repertoire. TC-mAb mice not only provide a valuable platform to obtain fully human therapeutic Abs but also a model to elucidate human Ig repertoire formation. Our findings may facilitate advancement of human Ig production research in the mouse.

## Results

**Constructing a novel IGHK-NAC containing fully human Ab genes.** To produce fully human Ab producing mice, sequential translocation cloning of human IGK (on hChr. 2) and IGH (on hChr. 14) loci into the MAC vector was conducted using Cre/loxP and FLP/FRT systems[9] (Fig. 1a and Supplementary Fig. 1). The MAC is composed of a native mouse centromere, a loxP site, part of the 3′ region of the hypoxanthine-guanine phosphoribosyl transferase (HPRT) gene, and telomeres. In addition, the MAC contained enhanced green fluorescent protein (EGFP) and neomycin resistance genes, driven by the ubiquitous promoters, CAG and PGK, respectively[9].

After insertion of the loxP and FRT sites proximally and distally of the IGK locus on hChr.2p, respectively, the modified hChr.2 was transferred into CHO cells carrying the MAC using microcell-mediated chromosome transfer (MMCT)[10]. An intended reciprocal translocation between the MAC and the modified hChr.2 by Cre/loxP recombination caused reconstitution of the HPRT gene and HAT resistance, which enabled us to select CHO cell lines carrying the MAC with the IGK locus (IGK-NAC). Using the same procedure, the distal region covering the IGH locus on hChr.14 was sequentially translocated into the IGK-NAC to produce CHO cells with the MAC carrying both IGK and IGH loci (IGHK-NAC) as described in the Methods section. IGHK-NAC construction was confirmed by genomic PCR analysis and fluorescence in situ hybridization (FISH) analysis (Supplementary Figs. 2–5).

**Generating a humanized trans-chromosomic (Tc) mouse.** To generate a humanized Tc mouse, the IGHK-NAC was transferred from CHO cells to mouse embryonic stem (ES) cells (TT2F: 39, XO) and mouse ES cells, in which endogenous Igk and Igh had been previously knocked out (6TG-9-mES: 39, XO)[11,13]. The mES hybrids with the desired karyotype containing the IGHK-NAC (40, XO, +IGHK-NAC) were isolated (Supplementary Fig. 6) and utilized to produce chimeric Tc mice. Mice with high coat colour chimerism were mated with endogenous Ig gene-knockout mice. After confirming the IGHK-NAC was transmitted through the germline, F1 mice were further mated with Igk- and Igh-KO mice carrying a Ig lambda light chain locus (Igλ) low allele[11] from CD-1 (ICR) mice to establish Tc mice carrying the IGHK-NAC with mouse Igk- and Igh-KO, and Igλ low background (designated as TC-mAb mice). Therefore, the TC-mAb mice generated show outbred strains and have a mixed genetic background derived from the ICR strain.

The retention rate of the IGHK-NAC in various tissues of the TC-mAb mice was evaluated by FISH analysis and by monitoring EGFP expression. This indicated a high percentage of IGHK-NAC retention (Fig. 1b–d). Notably, flow cytometry analyses revealed that the IGHK-NAC was also stable in blood cells, as reported previously[17]. The average percentage of EGFP-positive peripheral blood mononuclear cells (PBMCs) in F9–20 generation TC-mAb mice was 95.4% (Fig. 1e and Supplementary Fig. 7), confirming high stability of the IGHK-NAC over multiple generations, especially in blood cells. The average percentage of both EGFP- and B220-positive cells was 9.5% in TC-mAb mice (Fig. 1f and Supplementary Fig. 7), indicating the IGHK-NAC-dependent rescue of B cell production in the Ig-knockout

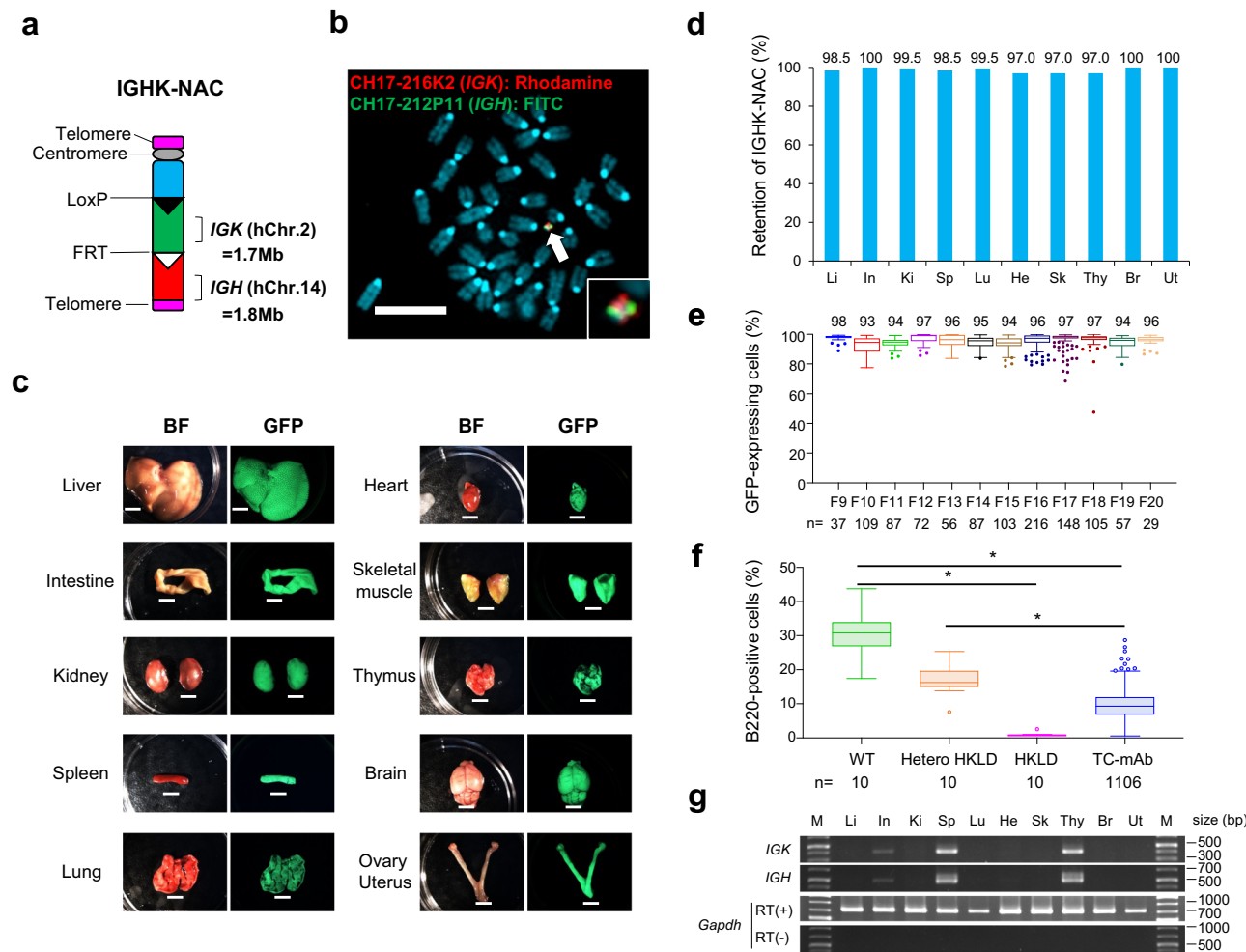

**Fig. 1 Generation of fully human Ab-producing mice. a** Representation of the IGHK-NAC structure. Human Ig heavy chain locus (*IGH*) derived from hChr.14 and Ig kappa light chain locus (*IGK*) derived from hChr.2 are represented on a mouse artificial chromosome (MAC). **b** Representative image of metaphase FISH analysis with IGK-BAC (CH17-216K2) (red) and IGH-BAC (CH17-212P11) (green) detecting the IGHK-NAC in bone marrow cells from a TC-mAb mouse. Arrow indicates the IGHK-NAC and the inset shows an enlarged image thereof. Scale bar (10 µm). **c** GFP images of different tissues from a Tc mouse carrying the IGHK-NAC. GFP expression indicates the presence of the IGHK-NAC. BF, bright field. Scale bar (5 mm). **d** Retention rate of the IGHK-NAC in various Tc mouse tissues analysed by FISH. Li liver, In intestine, Ki kidney, Sp spleen, Lu lung, He heart, Sk skeletal muscle, Thy thymus, Br brain, Ut uterus. **e** Percentage of GFP-expressing cells in peripheral blood mononuclear cells from Tc mice of different generations. **f** Percentage of B220-positive cells in the lymphocyte fraction of peripheral blood in age-matched WT ($n = 10$), hetero HKLD ($n = 10$, $Igh^{+/-}$, $Igk^{+/-}$ and $Igl1^{+/low}$), HKLD ($n = 10$, $Igh^{-/-}$, $Igk^{-/-}$ and $Igl1^{low/low}$) and TC-mAb mice ($n = 1106$). Box plots are indicated in terms of minima, maxima, centre, bounds of box and whiskers (1.5 interquartile range value), and percentile in the style of Tukey (**e**, **f**). *$P < 0.001$ (unpaired Dunn's test for multiple comparison). Gating strategies are presented in Supplementary Fig. 7. **g** RT-PCR analysis of total RNA from various tissues of the Tc mouse. Li liver, In intestine, Ki kidney, Sp spleen, Lu lung, He heart, Sk skeletal muscle, Thy thymus, Br brain, Ut uterus. M size marker. Twelve-week-old (**b**) and 14-week-old female TC-mAb mice (**c**, **d**, **g**) were analysed. Assessment of the GFP- and B220-positive ratios in lymphocytes by FCM was applied to 4–5-week-old TC-mAb mice (**e**, **f**). Five female and five male mice aged 4–5 weeks each were used for WT, hetero HKLD and HKLD (**f**).

mice (HKLD). It should be noted that the number of circulating B220-positive cells in TC-mAb mice with a single IGHK-NAC copy was 50–60% of that observed in double heterozygous mice (hetero HKLD) carrying one copy each of functional mouse Ig loci.

Gene expression from the IGHK-NAC was analysed by RT-PCR using total RNA from various tissues of the TC-mAb mice. Transcripts of human *IGK* and *IGH* were abundant in the thymus, spleen, and at low levels in the intestine (Fig. 1g), where B cells are related to T cell development, B cell maturation and IgA antibody production, respectively. Altogether, we successfully generated a strain of humanized IGHK Tc mice that displayed tissue-specific expression of the human Ig-genes on the IGHK-NAC.

**Analysis of human Ig heavy chain repertoire in TC-mAb mice.** Immunoglobulin genes are assembled by germline-encoded gene segment recombination: variable (V), diversity (D) and joining (J) for heavy chains, and V and J for light chains. Recent advances in next-generation sequencing (NGS) technologies enable comprehensive profiling of human antibody repertoires, including V(D)J gene usage in combinatorial rearrangement, junctional diversity, and SHM at an unprecedented scale. However, reported repertoire analyses in human Ig YAC transgenic mice producing fully human Abs have involved a limited number (~500) of cloned Ig sequences[18]. In this study, to investigate the transcript sequences of human Ig from IGHK-NAC, we employed NGS to interrogate human Ig repertoires in RNA samples extracted from spleen cells of five unimmunized TC-mAb mice, one OVA-immunized

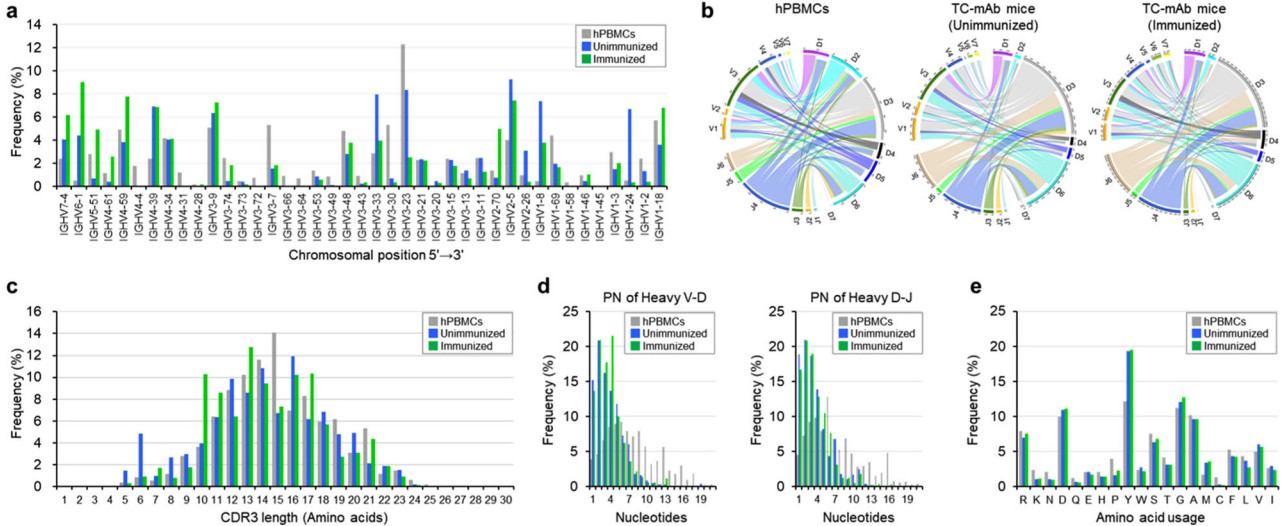

**Fig. 2 Repertoire analyses of heavy chain variable regions in TC-mAb mice. a** Human heavy chain gene utilization in hPBMCs from healthy donors and in TC-mAb mice with (female, 24 weeks old) and without (five female mice, 18 weeks old) immunization. Percentage frequency use of V gene segments in healthy human donors (grey), TC-mAb mice with (green) and without immunization (blue) are represented. **b** Circos plots comparing VDJ gene association. The gene segments are grouped as subfamilies and are shown together with the first digit of their allele name. Links indicated the relative frequencies of specific VDJ combinations, and wider links indicate higher frequencies of recombination. The relative CDRH3 length (**c**), P and N nucleotide addition at joining positions (**d**), and amino acid usage (**e**) within the CDRH3 of productive rearrangements.

TC-mAb mouse, and human PBMCs from healthy adult donors. Over 1.3 million qualified reads were accumulated from each sample and assembled into merged reads. IgBlast-annotated reads were collated into data sets for subsequent analyses (Supplementary Table 1). The saturation of clonotype variations was confirmed in the rarefaction curve of each sample. In addition, the annotated reads showed that the *IGH* and *IGK* transcripts in TC-mAb mice and in hPBMCs were similar and >92% productive. Comparison of the use of heavy chain V segments denoted as functional by the IMGT database[19,20] (Fig. 2a and Supplementary Table 2) showed that all of the functional V segments (41 segments) were detected in hPBMCs, unimmunized TC-mAb and immunized TC-mAb mice, and intriguingly, the broad distribution of V segment usage was very similar among these samples. For instance, of 20 V segments represented at a frequency of over 2% in hPBMCs, 17 were also detected at a frequency of over 2% in unimmunized TC-mAb or immunized TC-mAb mice. Additionally, in contrast to mice with human Ig YAC transgenes[18] and humanized VH regions[7], which exhibited biased use of the top three frequently used V segments (over 40%), use of the top three V segments in unimmunized TC-mAb mice was lower than 26%, as observed in PBMCs. The use of D segments was also mostly similar between TC-mAb mice and hPBMCs. *IGHD3–10* gene segments were the most frequently used segment in both hPBMCs and TC-mAb mice, although their use in TC-mAb mice (33.46%) was significantly higher than that in hPBMCs (11.79%) (Supplementary Table 2). No evidence existed of preferred recombination between D and J segments that are in close proximity, which was observed in human Ig YAC transgenic mice[3]. All six J segments were used, and the J4 and J6 gene segments were dominantly used in TC-mAb mice and hPBMCs.

Circos plots of V, D and J segment combinations in rearranged Ig heavy chains revealed that the frequency of V/D/J use and the combinations of VH regions were comparable between hPBMCs and TC-mAb mice (Fig. 2b and Supplementary Figs. 8–10), indicating no significant difference in V/D/J gene use and combination in TC-mAb mice with or without immunization. The Shannon-Weaver index, which indicates robust estimates of overall immune-repertoire diversity, was calculated using Eq. (1)

and also indicated that V(D)J recombination in TC-mAb mice occurred as efficiently as it does in humans and that human-like combinatorial diversity is formed in TC-mAb mice by common recombination mechanisms that rely on human-derived *cis*-regulatory elements in the *IGH* locus (Supplementary Table 3). These results suggested that carrying the entire human Ig locus of *IGH* was necessary and sufficient to faithfully reproduce the human Ab rearrangement process in the mouse.

**CDRH3 length distribution and amino acid composition.** Complementary-determining region 3 of the heavy chain (CDRH3) is located at the V/D/J junctional region of the Ig heavy chain; therefore, it is highly diverse and a key determinant of specificity in antigen recognition[21]. In hPBMCs and unimmunized TC-mAb mice, the distribution of CDRH3 length closely overlapped and the average length was 14.97 and 14.22 amino acids, respectively, with a broad distribution between 5 and 25 amino acids (Fig. 2c). Nucleic acid addition at V-D and D-J junctions was observed in the shorter distributions in TC-mAb mice compared with hPBMCs (Fig. 2d), which was consistent with a previous report indicating a significant decrease in the number of N additions in Ig YAC transgenic mice carrying part of both *IGH* and *IGK* loci[18]. The amino acid composition of CDRH3 among hPBMCs and TC-mAb mice was almost the same indicating that Ig-gene rearrangement was processed in the human manner (Fig. 2e).

**Repertoire analysis of human *IGK* chains in TC-mAb mice.** The human kappa locus has large, duplicated clusters of kappa chain variable (VK) segments (J-proximal and J-distal), which are separated by 800 kb. They should contribute to expanding the potential *IGK* repertoire. The *IGKV* distal cluster spans 400 kb and comprises 36 (16 potentially functional) V segments and the proximal cluster spans 600 kb and comprises 40 (18 potentially functional) V segments. Because of this complex structure, introducing the entire human *IGK* locus into mice has only been accomplished using our chromosome vector system. This enables evaluation of the *IGK* repertoire, which relies on the native

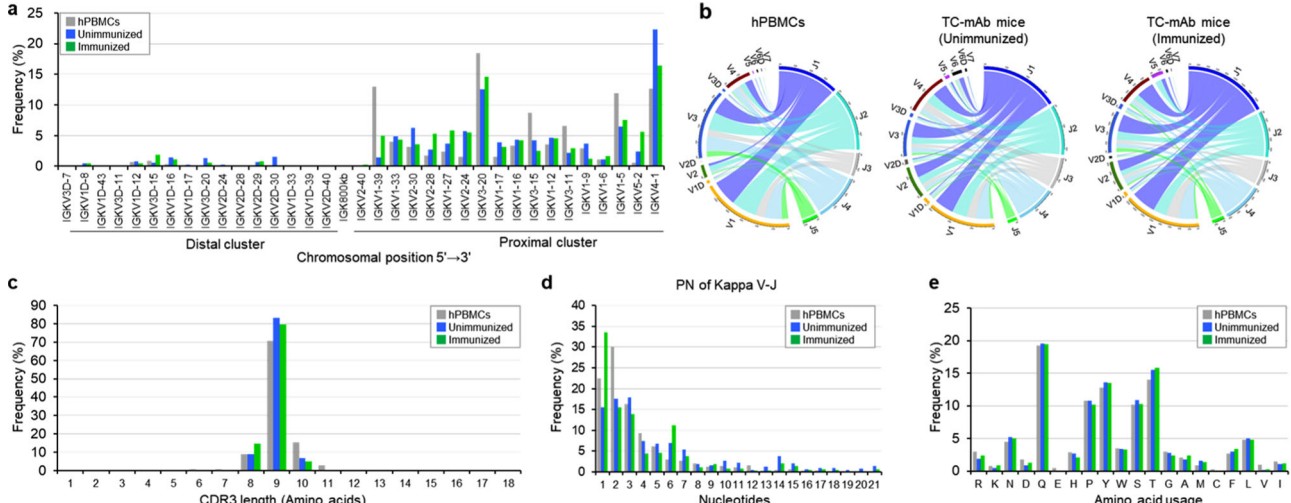

**Fig. 3 Repertoire analyses of light chain variable regions in TC-mAb mice. a** Human VK gene utilization in hPBMCs of healthy donors and in TC-mAb mice with and without immunization (same as Fig. 2). **b** Circos plots comparing VJ gene association. Links indicate the relative frequencies of specific VJ combinations. The relative CDRL3 length (**c**), P and N nucleotide addition at joining positions (**d**), and amino acid usage (**e**) within the CDRL3 of productive rearrangements.

configuration of entire human *IGK* locus. The NGS data indicated that all of the 18 potentially functional proximal V segments were used in both hPBMCs and TC-mAb mice (Fig. 3a). Eleven and 9 of 16 functional distal V segments were also detected in TC-mAb mice and hPBMCs, respectively, though their use was less frequent compared with proximal V segments. These results contrast well with previous data showing highly biased use of V segments in mice with a partial *IGK* YAC transgene[18] or a humanized VK region[5,7]. The Circos plots of V-J segments showed that the frequency use and assembled combinations of the VK region were highly similar between hPBMCs and TC-mAb mice (Fig. 3b and Supplementary Table 2). Additionally, no significant difference existed in J segment use in TC-mAb mice with or without immunization. Furthermore, CDRL3 length, the number of added nucleotides, and amino acid composition were all comparable between hPBMCs and TC-mAb mice (Fig. 3c–e). Altogether, as seen with the human Ig heavy chain repertoire, the formation of human-like combinatorial and junctional diversity was achieved in TC-mAb mice carrying a MAC with the entire human Ig kappa locus.

**Somatic hypermutation in TC-mAb mice.** SHM is a process in which point mutations accumulate in the variable regions of both the Ig heavy and light chain genes during B cell maturation, thereby enabling the selection of B cells producing high-affinity Abs against immunogens. On the basis of NGS data of the variable regions of heavy (VH) and kappa (VK) sequences, clone lineages for both IgM and IgG were identified to estimate the B cell receptor (BCR) repertoire diversification. The definitions of the clone lineage and clonotype are indicated in the Methods section. To detect the position of SHMs leading to amino acid changes in the VH and VK sequences, annotated reads were obtained from unimmunized and immunized TC-mAb mice, and were compared with their germline sequences.

The 70 most frequently used clone lineages of VH (CLH001–070) and the 20 most frequently used VK (CLL001–020) sequences were analysed and compared between immunized and unimmunized TC-mAb mice (Fig. 4a–d). The SHM in these clone lineages was accumulated in the CDR1, 2, 3, and FR3 regions of both VH and VK sequences. The mutations appeared more frequently in immunized TC-mAb mice, indicating selection with antigen administration. The

data of the top 20 clone lineages (CLH001–020 of VH and CLL001–020 of VK) were summarized for the fold change of annotated reads, the average length of CDR3, and the percentage of SHM with at least two mutations in a variable region (Supplementary Table 4). The fold change in annotated reads indicated that specific clone lineages of both VH and VK sequences were expanded during immunization, while the fold change of the VH sequence was more prominent than that of the VK sequence. However, a higher mutation rate in the VK sequence was observed compared with the VH sequence after immunization.

Phylogenetic trees (circular dendrograms) were drawn to show the mutation mapping and expansion of clonotypes based on their amino acid sequence in each clone lineage. On the basis of the number of copies with more than 100 reads in the OVA-immunized TC-mAb mice, 260 clonal lineages of the VH sequence and 148 lineages of the VK sequence were analysed. The circular dendrogram of the 10 most frequent clonal lineages of the VH sequence and VK sequence is presented in Fig. 4e and Supplementary Fig. 11. The sequence diversity of a clone, CLH001, is presented in Fig. 4f. The average fold-expansion among the top 20 clone lineages was 16.0 for VH sequence and 5.7 for VK sequence (Supplementary Table 4). These results indicated that expansion of clonotypes with SHM was induced by immunization. Taken together, immunizing TC-mAb mice results in B cell clonal selection and expansion followed by SHM, leading to antigen-driven VH and VK sequence diversification, which should contribute to the generation of B cells bearing higher-affinity antibodies.

**Serum expression of human Igs in TC-mAb mice.** Unimmunized TC-mAb mice (n = 17) were examined by enzyme-linked immunosorbent assays (ELISAs) to determine serum concentrations of human and mouse Ig proteins. The average levels of human Ig μ, γ and κ were 518.1, 153.2 and 688.6 μg/ml, respectively (Fig. 5a). As expected, the levels of mouse Ig μ, γ, κ and λ in unimmunized TC-mAb mice were at significantly lower levels, as reported in double Tc mice[11], compared with the parental WT mice (n = 12) in which the levels of Ig μ, γ and κ were 248.0, 993.4 and 637.0 μg/ml, respectively (Supplementary Table 5). Although we used the λ1 low allele[22] instead of Ig λ-knockout to reduce mouse Ig λ expression in TC-mAb mice,

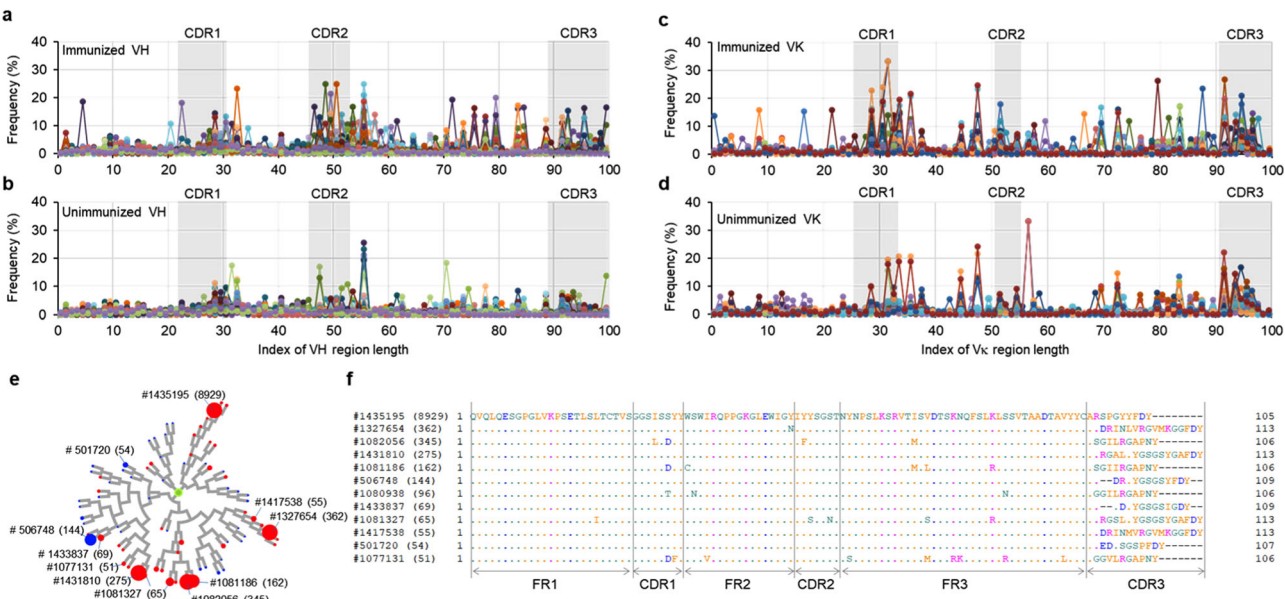

**Fig. 4 Somatic hypermutation analysis.** The frequency of SHM at every position of the variable region was calculated as described in the Methods section and is represented for VH and VK chains in OVA-immunized TC-mAb mice (**a**, **b**) and unimmunized TC-mAb mice (**c**, **d**) (same as Fig. 2). **e** The phylogenetic tree of the clone type CLH001. The circular dendrograms of the 20 most frequently used clone lineages were over-laid with the number of copies carrying the same CDRH3 sequences. The leaves of circular dendrograms have a maximum of 50 reads; therefore, as the number increases, the circle in the leaves increases. The colour of the circle indicates with (red) or without (blue) immunization in TC-mAb mice. The amino acid sequences of 12 clonotypes containing more than 50 copies in the CLH001 clone lineage. The clonotype numbers and each copy number are also indicated. **f** Amino acid sequence alignment of the 12 clonotype variable regions of CLH001.

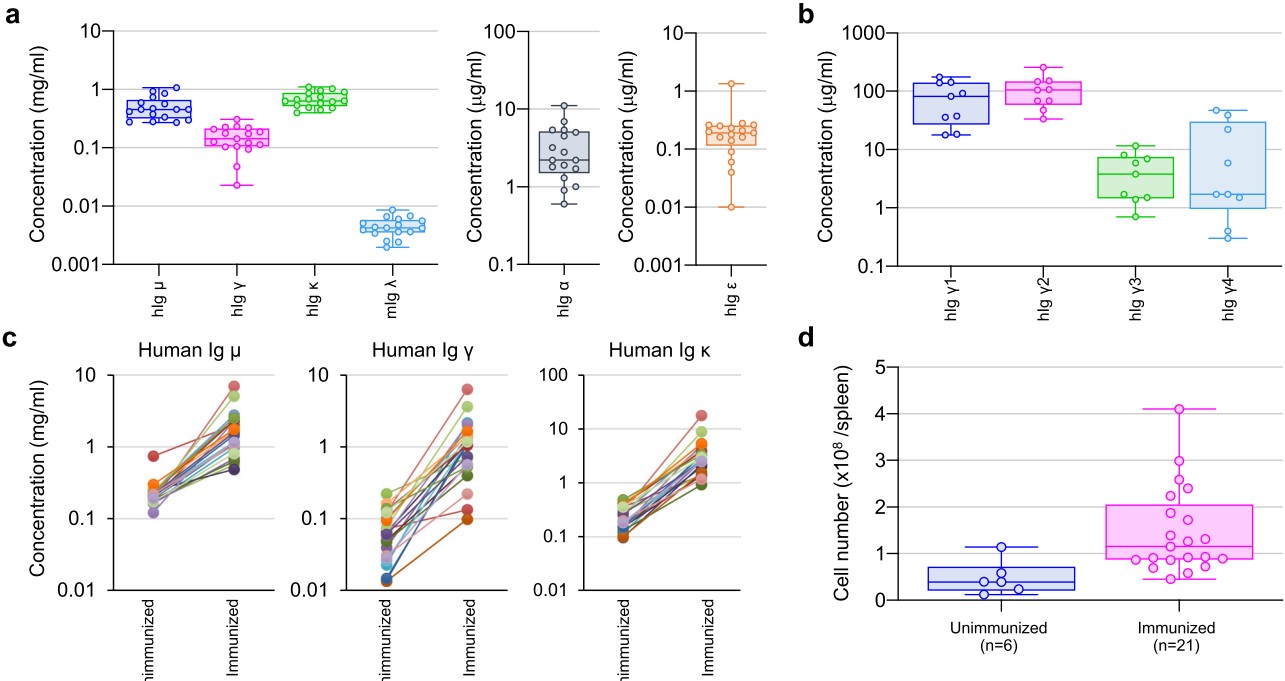

**Fig. 5 Production of human Ig in TC-mAb mice. a** The serum concentration of different classes of human Igs in 15-week-old TC-mAb mice. The concentration of hIg μ, hIg γ, hIg κ, mIg λ, hIg α and hIg ε in TC-mAb mice (n = 17) was measured. **b** The serum concentration of human IgG subclasses. The concentrations of Ig γ1, 2, 3 and 4 were measured in 15-week-old TC-mAb mice (n = 9). **a**, **b** Each symbol represents one serum concentration. **c** The serum concentration of hIg μ, Ig γ and Ig κ before and after immunization of TC-mAb mice. The symbols collected from the same mouse before and after immunization are connected by a line. **d** The number of lymphocytes. The numbers of harvested cells were counted under Turk's solution. Each symbol indicates one mouse. Box plots are indicated in terms of minima, maxima, centre, bounds of box and whiskers, and percentile.

the result indicates that the human IGK locus of the IGHK-NAC can compete well with the mouse λ1 low allele. The observation that levels of Ig γ were lower than those of Ig μ is consistent with data from double-Tc mice and Ig YAC Tg mice, which rely on human-derived constant region sequences[3,11]. All four γ subclasses (γ1, γ2, γ3 and γ4) were detected and the average concentrations were 82.2, 109.3, 4.6 and 13.3 μg/ml, respectively, and Ig α and Ig ε were also detectable (Fig. 5a, b, and Supplementary Table 5), which indicated production of all human Ig subclasses.

When TC-mAb mice (n = 21) were administrated with various antigens, including OVA, recombinant proteins expressed in *Escherichia coli*, and whole mammalian cell extract, the Ig μ, γ and κ levels and the number of spleen cells were greatly elevated (Fig. 5c, d). In particular, the fold change in the average level of Ig γ (8.6) was greater than that of Ig μ (4.0), which indicates a robust induction of class-switching from Ig μ to Ig γ.

**Producing antigen-specific fully human Abs in TC-mAb mice.** To optimize the immunization conditions for producing antigen-specific human Abs in TC-mAb mice, we first immunized age-matched TC-mAb (n = 2) and WT (ICR) mice (n = 2) with OVA using Freund's complete adjuvant. The OVA-specific human Ig γ response in TC-mAb mice was relatively delayed compared with that in WT mice (Fig. 6a, b and Supplementary Fig. 12), indicating the requirement for two or three additional booster steps to reach a plateau and similar OVA-specific mouse Ig titres to those in WT mice. Then, Trx-EpEX and GST-EpEX, each of which is a fusion protein containing the extracellular domain from human EpCAM, was used as an immunogen and for verifying the serum titre of EpEX-specific Abs, respectively. Trx-EpEX is a fusion protein containing thioredoxin 1 (Trx) as a Tag sequence. GST-EpEX contained glutathione S-transferase (GST) as an alternative to Trx. Following administration of Trx-EpEX in TC-mAb mice (n = 2), the titres of Trx-EpEX-specific human Abs increased linearly until the second or third booster steps and then increased slightly until the final booster step (Fig. 6c). The titres of GST-EpEX-specific human Ig were substantially increased after the third or fourth booster steps. After the fourth (individual A) and seventh boosters (individual B), each immunized TC-mAb mouse was euthanized and lymphocytes were harvested from the spleen and lymph nodes, respectively, for the production of EpEX-specific human mAbs (Supplementary Table 6). The human Ig isotypes of these EpEX-specific human mAbs (80 and 297 clones from individual A and B, respectively) were analysed (Fig. 6d) to validate class-switching in TC-mAb mice. While the majority (73.8%) of hybridoma clones from individual A (four boosters) produced IgM, those from individual B (seven boosters) produced IgG as the major isotype (86.9%). Further analysis showed that the percentages of IgG1, IgG2, IgG3 and IgG4 subclasses in IgG-producing hybridomas were 64.3%, 4.7%, 16.5% and 1.4%, respectively. It should be noted that the number of mAbs obtained from TC-mAb mice (50 clones) was much higher than that produced from Balb/c mice (8 clones) using the similar protocols (Supplementary Table 6). To analyse IGHK-NAC stability in hybridoma cells, native antigen-reactive clones were selected and established by limiting dilution cloning. The cloning success rate was over 96.1% (Supplementary Table 6), revealing that the IGHK-NAC was stable in hybridoma cells. We then measured total Ig levels in the serum of TC-mAb individual B (which received seven booster administrations) (Fig. 6e). In agreement with the increase in the titre of antigen Trx-EpEX-specific Ab (Fig. 6c), the serum concentrations of human Ig μ, γ and κ were robustly elevated after the second booster step. The human Ig μ level reached a plateau after the second booster step and the human Ig γ level exceeded that of human Ig μ after the

fifth booster step, which is consistent with the majority of antigen-specific mAbs isolated from this TC-mAb mouse (individual B) consisting of different IgG subclasses. Whereas in anti-AMIGO2 mAb production, titres of GST-AMIGO2-Ig-specific human Ig were rapidly increased, reaching a plateau after the second booster step, and the fourth booster was sufficient to induce IgG subclass (Supplementary Fig. 13).

Taken together, antigen-specific human Ab-producing hybridomas were generated with high efficiency in TC-mAb mice. However, depending on the antigen, two or more booster immunizations may be necessary to obtain the optimal response. These results also suggest that high population subsets of antigen-specific B cells are contained in the spleen of immunized TC-mAb mice.

**Characteristics of mAbs obtained from TC-mAb mice.** Analysis of amino acid sequences of mAbs obtained from TC-mAb mice indicated a high degree of humanness, which was determined using T20 score analyser[23]. This confirmed that hybridoma clones carried fully human Ig heavy and kappa chains (Supplementary Fig. 14). The affinity of human mAbs against EpCAM and AMIGO2 was estimated using surface plasmon resonance. The $K_D$ values of human mAbs have a substantial affinity against target proteins in the nanomolar range (Supplementary Table 7), which suggested TC-mAb mice provided high-affinity fully human mAbs.

**B cell development and antigen-specific B cells in TC-mAb mice.** The B cell lineages in bone marrow, spleen, lymph nodes and peritoneal exudate cells (PECs) of TC-mAb mice were analysed by flow cytometry using combinations of Abs against cell surface markers (Supplementary Table 8)[24,25] and compared among the age-matched TC-mAb and WT mice. The subset populations of Pro-B and Pre-B, and recirculating B cells in the bone marrow were indistinguishable but that of immature B cells was decreased by half in TC-mAb mice compared with WT mice (Fig. 7a–c, Supplementary Figs. 15 and 16 and Supplementary Tables 9 and 10). The number of lymphocytes in the spleens of TC-mAb mice was also decreased by half, whereas that in lymph nodes was higher in TC-mAb mice. The immature B cell percentages were decreased, and surprisingly, the IgD[hi] B subset was not detected (Supplementary Figs. 17–20). Analysis of transitional subsets revealed that the T1 and T3 populations were decreased while T2 was increased, which was possibly related to IgD expression[26–28], and led to unique B cell maturation in TC-mAb mice. The absolute number of follicular and marginal zone B cells was significantly reduced according to the reduced number of immature B cells in the spleens of TC-mAb mice (Fig. 7d, Supplementary Figs. 21 and 22 and Supplementary Table 10 and 11). The numbers of splenic B1a B cells and B1a/b and B2 B cells of PECs[29] did not differ significantly, but it appeared that fewer B2 cells were produced in TC-mAb than in WT mice (Fig. 7e, Supplementary Figs. 23–26 and Supplementary Table 11).

After primary immunization with OVA, the peak percentage of germinal centre (GC) B cell formation in B cells was increased and appeared to form later in TC-mAb than in WT mice (Fig. 8a–c, Supplementary Figs. 27–29 and Supplementary Table 12) and may be due to the delayed immune response of TC-mAb mice. The ratio of dark/light zone (DZ/LZ) in TC-mAb mice was different at least 7 days after immunization (7 weeks of age). Additionally, in vitro BCR stimulation with anti-human IgM Ab elicited Syk activation, and B cell-activating factor belonging to the tumour necrosis factor family (BAFF) treatment preserved the live B cells (Fig. 8d, e and Supplementary Figs. 30 and 31). The obtained human mAbs activated mouse FcγRIV

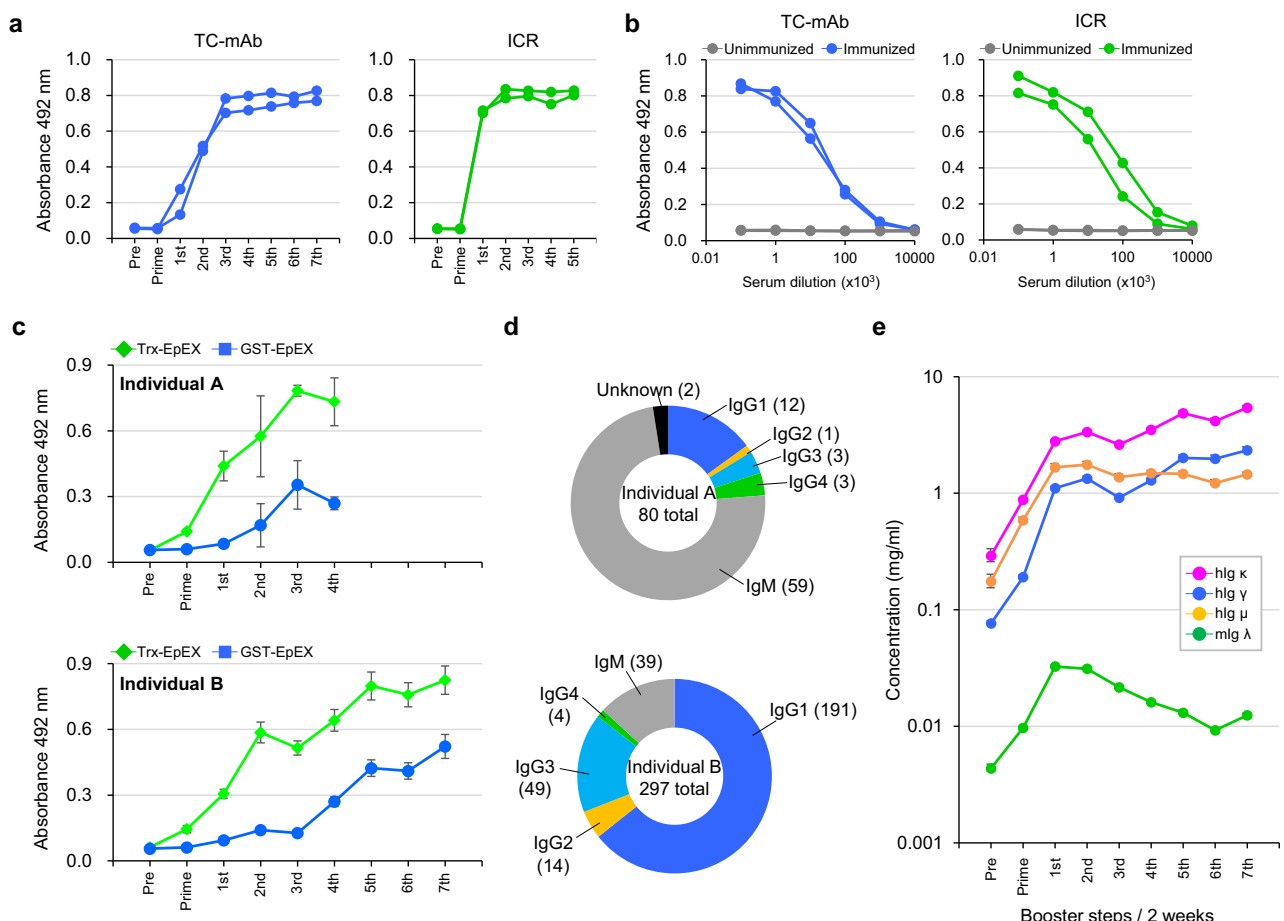

**Fig. 6 Immune response and production of monoclonal antibodies in TC-mAb mice. a** OVA-specific Ab titre. The titres of Abs against OVA were measured by ELISA using a species-matched secondary Ab. An anti-human IgG-Fc HRP conjugate was used for TC-mAb mice, and an anti-mouse IgG-Fc HRP conjugate was used for ICR. **b** Comparison of the anti-sera titres. The anti-sera samples were collected from TC-mAb mice after the seventh booster, and from ICR mice after the fourth booster. Samples were serially diluted by 1/10 from $10^2$ to $10^7$ and used to determine the OVA-specific titre by ELISA. **c** The antigen-specific titre in TC-mAb mice. Individual A (upper graph) was immunized with Trx-EpEX with the prime and four booster administrations. Individual B (lower graph) was immunized with the prime and seven booster administrations. The titres were analysed using the fusion proteins, Trx-EpEX (green) and GST-EpEX (blue). The Trx-EpEX titre indicated induction of antigen-specific Abs, but that of GST-EpEX indicated administration of EpEX-specific Abs. Each anti-sera was diluted at 8000-fold. Data are presented as means. Error bars indicate the standard deviation of triplicate measurements. **d** Determination of human Ig-classes. The mouse individual and the total number of class-determined clones are indicated in the centre of the circle. **e** The serum concentration of different classes of Ab. hIg μ (orange), hIg γ (blue), hIg κ (pink) and mIg λ (green) in the Trx-EpEX immunized TC-mAb mice are represented. Error bars indicate the standard deviation of triplicate measurements.

signalling (Supplementary Fig. 32). Allelic IgL exclusion clearly progressed between mouse Igλ and human Igκ, with a hIgκ:mIgλ ratio of 79.9:10.1 in the mature B cells of TC-mAb mice (Supplementary Figs. 33 and 34 and Supplementary Table 13). The sorted GC B cell repertoire indicated clonal expansion of some clonal lineages in VK (Supplementary Figs. 35 and 36). VH sequencing of the obtained anti-EpCAM mAbs indicated that the IGHV3–13 clone lineage was sequentially mutated to generate high-affinity mAbs (Supplementary Fig. 37). These data support that antigen-specific B cells were effectively produced in the spleen of TC-mAb mice.

In repeatedly immunized TC-mAb mice, we detected GC formation in the spleen (Fig. 9a), and the percentages of Ab-producing cells, including PB and PC, were significantly larger in TC-mAb than in WT mice (Fig. 9b, Supplementary Figs. 38 and 39 and Supplementary Table 14). A 4.7-fold increase (3.9% vs 0.8%) in the unimmunized state and a 2.7-fold increase (6.5% vs 2.4%) in the OVA-immunized state, respectively, existed. A high percentage of PB and PC subsets supported the efficient production of antigen-specific Abs in anti-sera of TC-mAb mice

(Fig. 6). Taken together, the higher ratios of PB and PC subsets and antigen-specific B cells in TC-mAb mice have desirable features for the production of antigen-specific therapeutic mAbs.

## Discussion

In this study, a new generation of fully human Ab producing mice (TC-mAb mice) was established that overcome the instability of human chromosome fragments in previously generated double Tc mice[11]. Our repertoire analysis of human Ab-producing mice by NGS revealed that the entire human heavy and kappa chain loci were used in the mouse similarly to those in hPBMCs, including the use of Ig-gene segments, combinations of V(D)J rearrangement, production of long-length CDRH3, and amino acid composition of CDRH3. Comparison with Ig-gene segment use in human/mouse chimeric Ab-producing mice (genetically chimeric mice) revealed that the Ig-gene repertoire was slightly or significantly different from that of hPBMCs[5,7,8,18] and TC-mAb mice, although a long CDRH3 length and similar DH and JH segments used in the human repertoire were already reproduced

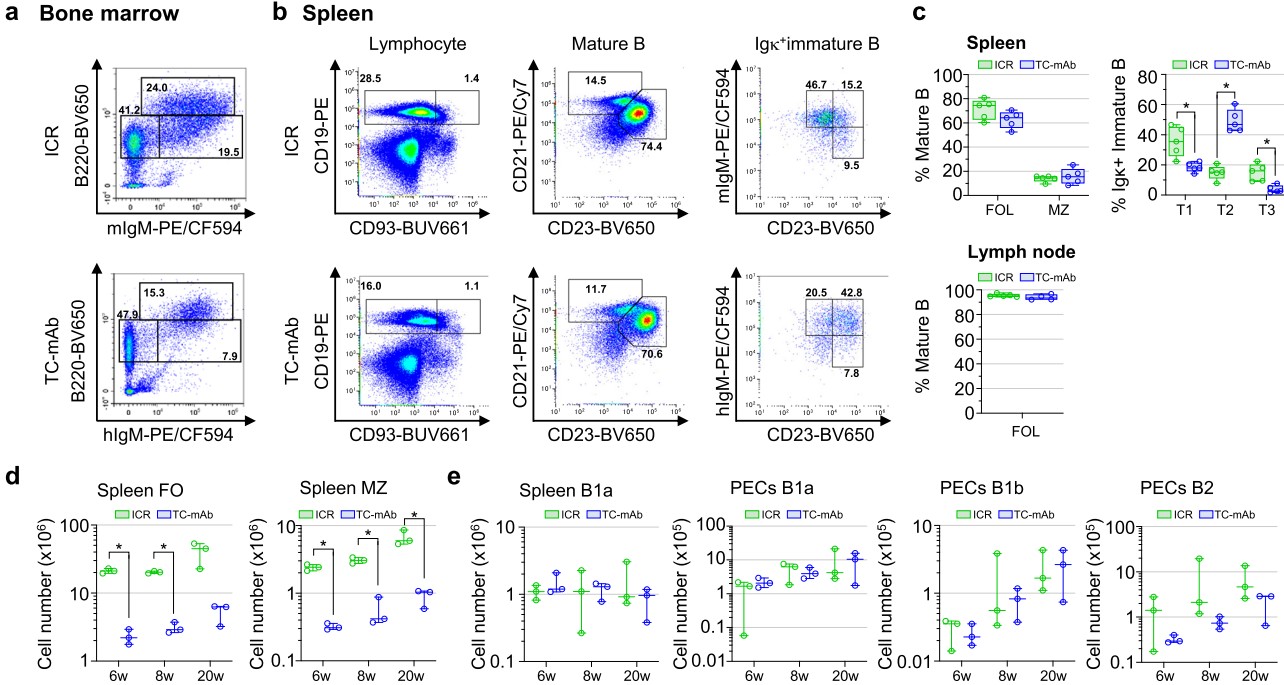

**Fig. 7 Analysis of B cell development. a** Flow cytometry of B cell subsets of early-developing B and mature B cell compartments in the bone marrow of 20–22-week-old ICR and TC-mAb mice. Panels indicate Pro-B/ Pre-B (IgM−B220+), immature B (IgM+B220+) and mature recirculating B (IgM$^{lo/+}$B220$^{hi}$) cells in bone marrow. Gating strategies are presented in Supplementary Fig. 15. **b** Flow cytometry of lymphocyte (left), mature (centre) and mouse/human Igκ+ immature B cell compartments. Gating strategies are presented in Supplementary Fig. 17. **c** The percentage of follicular (CD19+CD93−CD21+CD23+), and marginal zone (CD19+CD93−CD21$^{hi}$CD23−) B cells in the mature B cells and that of transitional 1 (IgM$^{hi}$CD23−), 2 (IgM$^{hi}$CD23+) and 3 (IgM$^{lo}$CD23+) B cells in the Igκ+ immature B cells of spleen (n = 5) and lymph node (n = 5/4 (ICR/TC-mAb)) are presented. Eight to 9-week-old unimmunized mice were analysed. All gating panels of B cell subsets are presented in Supplementary Figs. 18 and 20. P-values between ICR and TC-mAb mice of the percentages of B cell subsets were 0.115 (splenic follicular), 0.560 (marginal zone), 0.016 (transitional 1), <0.001 (transitional 2), 0.009 (transitional 3) and 0.301 (lymph nodus follicular). Box plots are indicated in terms of minima, maxima, centre, bounds of box and whiskers (1.5 interquartile range value), and percentile in the style of Tukey. *P < 0.05 (two-tailed unpaired Student's t test). The number of follicular and marginal zone B cells in the spleen mature B cells (**d**) and B1a (CD19+B220$^{lo}$CD5+CD43+) in spleen and B1a/b and B2 in PECs (**e**) at 6, 8 and 20 weeks-age of unimmunized mice are represented. All gating panels of B cell subsets are presented -n Supplementary Figs. 21–26. P-values of the cell number of B cell subsets between ICR and TC-mAb are indicated in the Source Data file. Data are presented as means. Error bars indicate the standard deviation of triplicate measurements. Results were pooled from two independent experiments. *P < 0.05 (two-tailed unpaired Student's t test).

in genetically chimeric mice[7]. In addition, a high frequency of SHM in CDR regions was detected in both VH and VK regions (Fig. 4a–d). Therefore, our analysis of TC-mAb mice strongly suggests that carrying the entire human Ig locus of *IGH* and *IGK* was necessary and sufficient to faithfully reproduce the human Ab rearrangement process in the mouse. These results are consistent with those of a previous report[30], indicating that the human gene expression pattern was recapitulated by the human trans-chromosome in mouse cells and that the genetic sequence was largely responsible for directing transcriptional programmes in homologous tissues. It is clear that "genomically" humanized animals, generated by Tc technology using mouse artificial chromosomes, are powerful tools to verify the function of genomic loci[9].

In contrast to genetically chimeric mice, which exhibit no major defects or differences in B cell differentiation compared with WT mice, some unique profiles of B cell development and Ab production were also observed in Tc-mAb mice. This is consistent with reports showing species incompatibility of Ig genes between humans and mice[3,11], which may be due to the interactions between the human Fc region and mouse receptors such as Fc-gamma receptors and Ig-alpha/Ig-beta heterodimer[3,11]. For instance, a lower concentration of IgG (Fig. 5a and Supplementary Table 5) and altered B cell development (Figs. 7a–e, 8a–c and Supplementary Tables 9–14) in TC-mAb mice compared with WT mice as well as the

requirement of two or three additional booster steps for some antigens to elicit an optimal immune response were observed (Fig. 6a). When a "putative threshold" that elicits immune response activation exists, TC-mAb mice probably have a higher threshold for some antigens compared with WT mice. Intriguingly, our detailed analyses showed high titre antigen-specific Abs containing all human subclasses (Fig. 5a, b), and a large number of antigen-specific Ab-producing hybridoma cells with SHM and affinity maturation (Fig. 4e and Supplementary Figs. 11, 36 and 37), suggesting that various antigen-specific human Abs were efficiently produced in TC-mAb mice. Because unimmunized TC-mAb mice showed low Ig γ concentration (Supplementary Table 5) and few IgG-positive subsets (Supplementary Table 14), an environment might have existed in which naïve B cells and antigen-specific memory B cells could easily respond to antigen and provide IgG-positive B cells by homeostatic proliferation[31,32] or other unknown mechanisms. This speculation is supported by the high production rate of target-specific mAbs from TC-mAb mice (Supplementary Fig. 13 and Supplementary Table 6), although further investigations into the interactions between human IgGs and mouse Fcγ receptors[33,34] are needed to understand the underlying mechanisms, which could have important implications for the development of high efficiency therapeutic mAb production. Because the TC-mAb mice produced in this study were established from an outbred strain, the detailed analyses above require the generation of animals backcrossed with

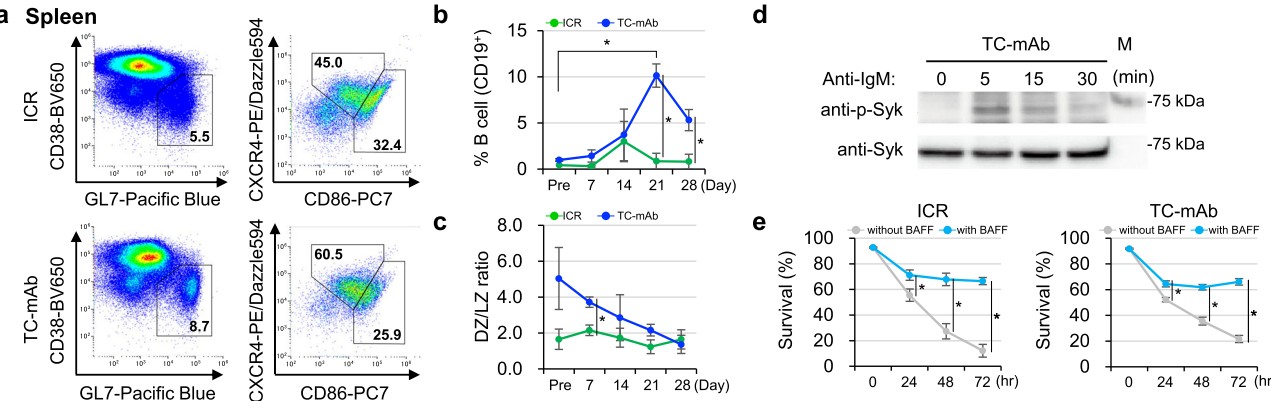

**Fig. 8 Analysis of spleen cells in immune system stimulation. a** Flow cytometry of germinal centre B cells (CD19+CD38lo/-GL7+) (left) and dark (CD19+CD38lo/-GL7+CXCR4hiCD86lo) and light zone (CD19+CD38lo/-GL7+CXCR4loCD86hi) (right). The percentage of germinal centre B, dark zone and light zone at day 14 after immunization are represented. Gating strategies are presented in Supplementary Fig. 27. **b** The percentage of cells in the germinal centre B cells in CD19+ B cells of ICR (green) and TC-mAb (blue) mice. All gating panels of germinal centre B cell subsets are presented in Supplementary Fig. 28. *P*-values (two-tailed unpaired Student's *t* test) between ICR and TC-mAb mice of the percentages of germinal centre B subsets were 0.202 (Pre, pre-immune), 0.139 (day 7), 0.790 (day 14), 0.002 (day 21) and 0.013 (day 28). *P*-values between pre and a peak timepoint of germinal centre B subset percentages were 0.236 (ICR, pre and day 14) and 0.009 (TC-mAb, pre and day 21). **c** The ratio of dark and right zone. All gating panels of germinal centre B cell subsets are presented in Supplementary Fig. 29. *P*-values between ICR and TC-mAb mice of the ratio of dark and light zone B subsets were 0.097 (Pre), 0.006 (day 7), 0.345 (day 14), 0.061 (day 21) and 0.626 (day b 28). **d** Phosphorylation status of Syk of B cells. **e** B cells from ICR and TC-mAb mice were cultured in the absence (grey) or presence (sky blue) of BAFF for 24, 48 or 72 h. Staining with TOPRO3 was analysed by flow cytometry and percentages of TOPRO3-negative gated cells (live cells) are shown. Data are presented as means. Error bars indicate the standard deviation of triplicate measurements. Results were pooled from two independent experiments. *P < 0.05 (two-tailed unpaired Student's *t* test).

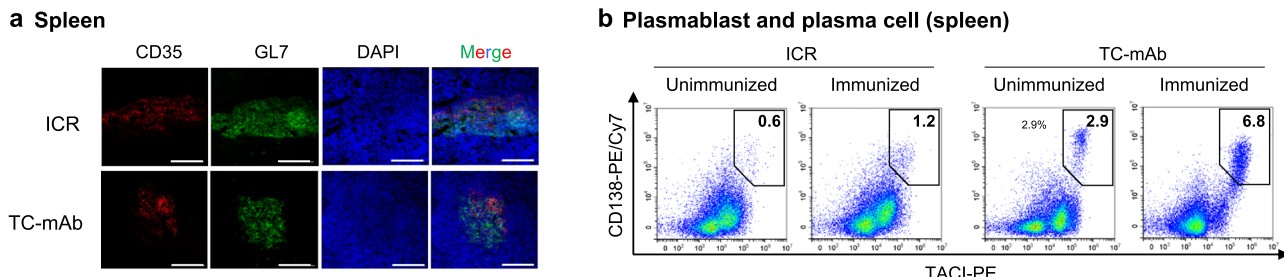

**Fig. 9 Analysis of spleen cells in repeatedly immunized mice for monoclonal antibody production. a** Representative images of fluorescence immunohistochemistry. The spleens of OVA-immunized TC-mAb mice were harvested and analysed by immunohistochemistry. Sections were stained for follicular dendritic cells (CD35, red), GC B cells (GL7; green) and nuclei (DAPI, blue). The images were taken at ×20 magnification. All scale bars indicate 100 μm. Representative figure from two independent experiments is shown. **b** Flow cytometry of plasmablasts/plasma cells (CD138+TACI+). Each spleen sample was collected after the seventh and final booster steps of immunization (in mice of ~21 weeks old). Gating strategies and all gating panels of plasmablast and plasma cell subsets are presented in Supplementary Figs. 38 and 39.

an inbred strain. Furthermore, the TC-mAb mice have only one allele of human *IGH* and *IGK* loci in the IGHK-NAC; therefore, inter-mating of TC-mAb mice to generate mice carrying two copies of IGHK-NAC may improve the immune response and lymphocyte production rate. Alternatively, swapping hIgG class switch regulator elements from human to mouse in the IGHK-NAC may also improve these responses[27].

In conclusion, we have summarized the main characteristics of TC-mAb mice in comparison with other fully human Ab-producing mice (Supplementary Table 15). Our results suggest that TC-mAb mice offer a valuable platform for obtaining fully human therapeutic Abs[35–38] and a useful model for elucidating that regulation of human Ig repertoire formation.

## Methods

**Materials**. CHO (*Hprt*−/−) cells (JCRB0218) were obtained from the Japanese Collection of Research Bioresources Cell Bank (Osaka, Japan). CHO K1 cells (RCB0285) and DT40 cells (RCB1464) were obtained from the Riken BioResource Research Center (Ibaraki, Japan). P3X63Ag8.653 myeloma cells (CRL-1580) and HCT116 cells (CCL-247) were purchased from the American Type Culture Collection (Manassas, VA, USA). TT2F cells were provided by RIKEN Centre for

Biosystems Dynamics Research (Hyogo, Japan). Restriction and DNA-modifying enzymes were purchased from New England Biolabs (Ipswich, MA, USA) and TOYOBO (Tokyo, Japan), respectively. Primers were obtained from Eurofins (Huntsville, AL, USA). *E. coli* strains DH5α and Rosseta-gami B(DE3) pLysS were purchased from Takara Bio (Shiga, Japan) and Merck Millipore (Burlington, MA, USA), respectively.

**Cell culture**. A MAC vector was used to generate IGHK-NAC[9]. The MAC contained a mouse centromere, EGFP flanked by HS4 insulators, PGK-neo, loxP site-3′-HPRT, PGK-puro and telomeres. Chicken DT40 cells containing hChr.2 or hChr.14 were maintained at 40 °C in RPMI 1640 medium supplemented with 10% foetal bovine serum (FBS), 1% chicken serum, 50 μM 2-mercaptoethanol and 1.5 mg/mL G418. *Hprt*-deficient Chinese hamster ovary [CHO (*Hprt*−/−)] and CHO K1 cells were maintained at 37 °C in Ham's F-12 nutrient mixture (Invitrogen, Carlsbad, CA, USA) supplemented with 10% FBS. CHO cells containing the IGHK-NAC were maintained in medium with 800 μg/mL G418. Mouse embryonic fibroblasts (MEFs) were isolated from embryos at 13.5 days postcoitum (d.p.c.). MEFs were grown in Dulbecco's modified Eagle's medium (DMEM) (Sigma-Aldrich, St. Louis, MO, USA) containing 10% FBS. Parental mouse ES cell line (TT2F), endogenous Ig KO ES cell subline (6TG-9) and microcell hybrid TT2F and 6TG-9 clones were maintained on mitomycin C (Sigma-Aldrich)-treated Jcl:ICR (CLEA Japan, Tokyo, Japan) MEFs and neomycin-resistant MEFs (Oriental Yeast Co., Ltd., Tokyo, Japan), respectively, as feeder layers in DMEM with 18% FBS (Hyclone Laboratories, Logan, UT, USA), 1 mM sodium pyruvate (Invitrogen),

0.1 mM non-essential amino acids (Invitrogen), 0.1 mM 2-mercaptoethanol (Sigma-Aldrich), 2 mM L-glutamine (Invitrogen) and 1000 U/mL leukaemia inhibitory factor (Funakoshi, Tokyo, Japan).

**Construction of targeting vectors.** Targeting vector to introduce loxP-5′-HPRT into hChr.2. To prepare the homology arm, DNA from DT40 cells with hChr.2 was used as a template for PCR. A 9.5 kb homology arm was amplified using the following primers: cos138-F6B and cos138-R6B. The fragment was cloned into the BamHI site of the pKO Scrambler V901 backbone vector (Lexicon Genetics, Woodlands, TX) (V901-cos138). A PGKHyg-loxP-5′-HPRT fragment obtained by AscI and KpnI digestion was cloned into the SpeI site of V901-cos138 by blunt-end ligation (pCos138HL5′-H).

Targeting vector to introduce FRT-5′-HPRT into hChr.2. A FRT-5′-HPRT unit with cloning sites was synthesized (pkD9FRT). A CMV-Bsd fragment was inserted into pkD9FRT using EcoRI and XhoI sites to construct pBsdkD9FRT. A 4.1 kb left arm was amplified using primers kD-R9La L and kD-R9La R, and cloned into pBsdkD9FRT using NotI and MluI sites. A 3.2 kb right arm was obtained by PCR using primers kD-F9 Ra L and kD-F9Ra R, and inserted into the BamHI site (pBkD9FLR).

Targeting vector to introducing FRT-3′-HPRT into hChr.14. DNA from DT40 cells with hChr.14 was used as a template for PCR to prepare the homology arm. An FRT site with cloning sites was synthesized (pSC355FRT). PGKhyg and 3′-HPRT fragments were inserted into pSC355FRT using KpnI/ClaI and NheI/MluI, respectively, to construct pSC355HF3′H. A 3.8 kb left arm was amplified with primers NotISC355-F and AscISC355-R, and cloned into the NotI/AscI sites of pSC355HF3′H to construct pSC355HF3′HL. A 4.2 kb right arm was then prepared using primers SalISC355-F4 and BamHISC355-R4, and subcloned into pSC355HF3′HL to construct pSC355HF3′HLR. The primer sequences are described in Supplementary Table 16.

**FISH.** Trypsinized cells and homogenized tissue samples were incubated for 15 min in 0.075 M KCl, fixed with methanol and acetic acid (3:1), and then slides were prepared using standard methods. FISH analyses were performed using fixed metaphase or interphase spreads of each cell hybrid using digoxigenin-labelled (Roche, Basel, Switzerland) DNA [human COT-1 DNA/mouse COT-1 DNA (Invitrogen), mouse minor satellite DNA and IGK-BAC (CH17-405H5 and CH17-216K2)] and biotin-labelled DNA [human COT-1 DNA/mouse COT-1 DNA, IGK-BAC (CH17-140P2), IGH-BAC (CH17-262H11, CH17-212P11 and RP11-731F5) and each part of the targeting vector], essentially as described previously[10]. Chromosomal DNA was counterstained with DAPI (Sigma-Aldrich). Images were captured using an AxioImagerZ2 fluorescence microscope (Carl Zeiss GmbH, Jena, Germany).

**Modification of hChr.2 and hChr.14 in DT40 cells.** Homologous recombination-proficient chicken DT40 cells ($1 \times 10^7$) in 0.5 mL RPMI with 25 µg of linearized targeting vector were electroporated at 550 V and 25 µF using a Gene Pulser (Bio-Rad, Hercules, CA, USA). Drug-resistant DT40 clones were selected in 1.5 mg/mL G418, 10 µg/mL blasticidin S, or 1.5 mg/mL hygromycin. Homologous recombination in DT40 hybrid clones was identified by PCR using the primers described in Supplementary Table 16.

**Microcell-mediated chromosome transfer.** MMCT was performed as described previously[10]. hChr.2-loxPFRT and hChr.14-FRT in DT40 cells were transferred to CHO (MAC) and CHO (Hprt$^{-/-}$) cells, respectively, via MMCT. For each transfer, microcell hybrids were selected in medium with 800 µg/mL G418, 6 µg/mL blasticidin S, and 10 µM ouabain, and 300 µg/mL G418 and 10 µM ouabain, respectively. CHO IGK-NAC and IGHK-NAC were transferred to CHO (hChr.14-FRT) and CHO K1 cells, and selected with 600 µg/mL G418 and 4 µg/mL blasticidin S, and 800 µg/mL G418, respectively. To transfer IGHK-NAC to mouse ES cells, CHO K1 cells with IGHK-NAC were used as donor microcell hybrids. Briefly, mouse ES cells were fused with microcells prepared from donor hybrid cells and selected with G418 (250 µg/mL). The transferred IGHK-NAC in mouse ES cells was characterized by PCR and FISH.

**DNA transfection.** The Cre expression vector pBS185 (Invitrogen) or pCAG-FLPo was transfected into CHO hybrids with the MAC vector and modified hChr.2 or IGK-NAC and modified hChr.14 using Lipofectamine 2000 reagent (Invitrogen) in accordance with the manufacturer's protocol. After 24 h of culture in basic growth medium, the cells were cultured in medium with 1× HAT (Sigma) and 4–6 µg/mL blasticidin S for selection. Fourteen days later, drug-resistant colonies were picked up and expanded for further analyses.

**Genomic PCR.** Genomic DNA was extracted from cell lines and Tc mouse tissue specimens using a genomic extraction kit (Gentra System, Minneapolis, MN, USA). PCR was then performed using the primers listed in Supplementary Table 16. Primers for hChr.2 detection were D2S177 F/R, FABP1-F/R, EIF2AK3-F/R, RPIA-F/R, IGKC-F/R, IGKV-F/R, Vk3-2 F/R and D2S159_1 F/R. Primer pairs to detect the targeted hChr.2 were cos138 sp L PAGE/cos138 sp R, x6.1 cos RA L/R, kD9

tcLa L/R and kD9 tcRa L/R. Primer pairs for hChr.14 were MTA1-F3/R3, ELK2P2-F/R, g1(g2)-F/R, CH3F3/CH4R2, and VH3-F/R. Primer pairs to detect the targeted hChr.14 were 14TarC_La F/R and 14TarC_Ra F/R. Primer pairs to detect recombination junctions were KJneo/PGKr-2, TRANS L1/R1 and PGK-r2/CMVr-1. Mouse Igκ and Igh KO were confirmed by HKD mCk L1/R1 and mCk L1/R1, and HKD mCmu L1/R1 and mCmu L1/R1, respectively. Igλ low mutation was confirmed by PCR with mIglc1 VnC L/J3C1, followed by KpnI digestion. PCR was performed using AmpliTaq Gold (PerkinElmer, Waltham, MA, USA), KOD FX (TOYOBO), or AccuPrime Taq DNA polymerase (Invitrogen, Carlsbad, CA, USA). Amplified fragments were resolved by electrophoresis on 2% agarose gels, followed by staining with ethidium bromide.

**IGHK-NAC construction.** Construction of the MAC with human IGH and IGK loci employed PCR using the primers listed in Supplementary Table 16 and FISH at each step. Human chromosomes 2 and 14 were modified in homologous recombination-proficient chicken DT40 cells for recombination-mediated translocation. First, a loxP site was inserted proximally to the IGK locus on hChr.2p by homologous recombination. The targeting vector was introduced into DT40 cells with an intact hChr.2 and Neo resistance gene by electroporation and drug-resistant clones were obtained in medium with 1500 µg/mL hygromycin. FISH confirmed independent maintenance of a single copy of hChr.2 with the loxP unit (DT40 hChr.2loxP) (Supplementary Fig. 2a, b). Next, the FRT site was introduced distally to the IGK locus on hChr.2p. For loxP insertion, the targeting vector was introduced into DT40 cells with hChr.2loxP and drug-resistant clones were obtained in medium with 10 µg/mL blasticidin S. FISH confirmed independent maintenance of a single copy of hChr.2 with an FRT unit (DT40 hChr.2loxPFRT) (Supplementary Fig. 2c, d). The modified hChr.2 was transferred to CHO Hprt$^{-/-}$ cells with the MAC via MMCT[9]. Microcell hybrids were selected in medium with 800 µg/mL geneticin and 6 µg/mL blasticidin S. FISH revealed that the MAC and modified hChr.2 were independently and stably maintained in host CHO cells (Supplementary Fig. 3b).

Then, a distal region of hChr.2p from the loxP site, which included the IGK locus, was translocated to the MAC by Cre/loxP recombination (Supplementary Fig. 3a). A vector that expressed Cre under control of the CMV promoter (pBS185) was transfected by lipofection into CHO cells with the MAC and modified hChr.2. An intended reciprocal translocation between the MAC and modified hChr.2 by Cre/loxP recombination caused reconstitution of the HPRT gene in the by-product with HAT resistance, which enabled selection of CHO cell lines that carried the MAC with the IGK locus (IGK-NAC) and the by-product. Therefore, drug-resistant clones were obtained by selection in medium with 1× HAT and 4 µg/mL blasticidin S. Each recombination junction was detected by PCR and the structure of the IGK-NAC and by-product was confirmed by FISH (Supplementary Fig. 3b).

Next, we introduced a FRT site proximally to the IGH locus on hChr.14q in DT40 cells by homologous recombination (Supplementary Fig. 4a). We did not delete the distal side of the IGH locus because the IGH locus is located at the very end of hChr.14q. The targeting vector for FRT insertion was introduced by electroporation into DT40 cells that carried an intact hChr.14 with a Neo resistance gene. Drug-resistant clones were obtained by selection in medium with 1500 µg/mL hygromycin. FISH confirmed accurate targeting of hChr.14 in DT40 cells (DT40 hChr.14FRT) (Supplementary Fig. 4b). The modified hChr.14 was transferred from DT40 cells to CHO Hprt$^{-/-}$ cells via MMCT and microcell hybrids were obtained in selection medium with 300 µg/mL geneticin. FISH was used to confirm CHO cells with the modified hChr.14 (CHO hChr.14FRT) (Supplementary Fig. 4b). The IGK-NAC was then transferred to CHO hChr.14FRT cells by MMCT and microcell hybrids were obtained by selection in medium with 600 µg/mL geneticin and 6 µg/mL blasticidin S. FISH confirmed that the IGK-NAC and modified hChr.14 coexisted independently and stably in host CHO cells (Supplementary Fig. 5b). To clone the IGH locus into the IGK-NAC, FRT/FLP recombination-mediated reciprocal translocation between the IGK-NAC and modified hChr.14 was performed in CHO Hprt$^{-/-}$ cells and HPRT gene reconstruction with the desired product again enabled selection of CHO cells that carried the IGK-NAC with the IGH locus (IGHK-NAC) and the by-product. Drug-resistant clones were selected in medium with 1× HAT and 6 µg/mL blasticidin S. FISH revealed that the IGHK-NAC and by-product were independently and stably maintained in host CHO cells (Supplementary Fig. 5a, b). The resultant IGHK-NAC was transferred to CHO K1 cells to generate donor CHO K1 cells with a single desired chromosome, IGHK-NAC, for further MMCT. Microcell hybrids were selected in medium with 800 µg/mL geneticin and were monitored by GFP expression. FISH confirmed that a single copy of IGHK-NAC was independently maintained in host CHO K1 cells (Supplementary Fig. 6a, b).

**TC-mAb mouse generation.** To generate chimeric mice, mES cell lines were injected into eight-cell-stage embryos derived from ICR mice (CLEA, Tokyo, Japan) and then transferred into pseudopregnant ICR females. Chimeric mice with 100% coat colour chimerism were used for germline transmission. Chimeric mice were derived from an endogenous Ig KO mouse ES cell subline (6TG-9), C57BL/6 (female) × CBA (male) F1 genetic background with mouse Igh and Igκ KO, and were crossed with Jcl:ICR mice (CLEA Japan) with an ICR genetic background. F1 littermates were crossed each other or with Jcl:ICR mice. In subsequent generations, mice in the same generation were crossed with each other to produce and

maintain mice with mouse *Igh* and *Igκ* KO (HKD mice). F2 mice were obtained as described above and were crossed with Crl:CD1 mice (Charles River Laboratories Japan) that have the ICR/CD-1 genetic background with *Igλ* low allele(s). In subsequent generations, mice in the same generation were crossed each other or with HKD mice to produce and maintain mice with mouse *Igh* and *Igκ* KO, and *Igλ* low alleles (HKLD mice). Chimeric mice derived from 6TG-9 mES cells with IGHK-NAC were crossed with HKD mice and offspring were further crossed with HKLD mice to generate TC-mAb mice. TC-mAb mice were maintained by crossing TC-mAb and HKLD mice. Therefore, the TC-mAb mice generated were outbred strains with a mixed genetic background derived from the ICR strain. In this study, HKLD and TC-mAb mice were of more than 10 and six generations, respectively. Resultant TC-mAb mice were used in FISH, FCM, RT-PCR and several functional assays. Representative data from these assays are shown in each figure. Jcl:ICR (RRID:IMSR_JCL:JCL:mOT-0001) and BALB/cAJcl (RRID:IMSR_JCL:JCL:mIN-0005) mice were purchased from CLEA (Tokyo, Japan) and Crl:CD1 mice were purchased from Charles River (Kanagawa, Japan). The type of animal facility was specific pathogen-free (SPF), and experimental and control animals were cohoused in a controlled ambient temperature environment with a 12-h light/dark cycle. Mice underwent isoflurane-induced anaesthesia for all blood draws and other sampling. All animal experiments were approved by the Animal Care and Use Committee of Tottori University (Permit Numbers: 14-Y-23, 15-Y-31, 16-Y-20, 17-Y-28, 19-Y-22, 20-Y-13, 20-Y-31 and 21-Y-26).

**RT-PCR**. Total RNA from Tc tissue specimens was prepared using ISOGEN (Nippon Gene, Tokyo, Japan), treated with RNase-free DNase I (Wako Pure Chemicals, Osaka, Japan), and purified using RNeasy columns (Qiagen, Hilden, Germany), in accordance with the manufacturer's instructions. First-strand cDNA synthesis was performed using random hexamers and SuperScript III reverse transcriptase (Invitrogen). Primer pairs for the detection of human Ig-gene expression were as follows: Vk1BACK/Ck and CH4BACK/Cmu-1[11]. GAPDH (RPC1/2) was used as an internal control. The primer sequences for RT-PCR analyses are described in Supplementary Table 16. cDNAs from ICR tissues were used as negative controls. PCR was performed with cDNA using AmpliTaq Gold (PerkinElmer, Waltham, MA, USA). Amplified fragments were resolved by electrophoresis on 2% agarose gels, followed by staining with ethidium bromide.

**Deep sequencing analysis of Ab-coding transcripts**. An NGS analysis was performed using the unbiased TCR/BCR repertoire analysis technology developed by Repertoire Genesis (Osaka, Japan). In brief, unbiased adaptor-ligation PCR was performed as previously described[39]. Total RNA was converted to cDNA with Superscript III reverse transcriptase (Invitrogen) and the BSL-18E primer containing polyT18 and a *Not*I site. Following cDNA synthesis, double-stranded (ds)-cDNA was synthesized with *Escherichia coli* DNA polymerase I (Invitrogen), *E. coli* DNA ligase (Invitrogen) and RNase H (Invitrogen). The ds-cDNA was blunted with T4 DNA polymerase (Invitrogen). A P10EA/P20EA adaptor was ligated to the 5′ end of the ds-cDNA and then cut with a *Not*I restriction enzyme. After elimination of the adaptor and primer with a MinElute Reaction Cleanup Kit (Qiagen), PCR was performed with KAPA HiFi DNA polymerase (Kapa Biosystems, Wilmington, MA, USA) using an IgG constant region-specific primer CG1 for BCR and P20EA. The PCR conditions were as follows: 98 °C (20 s), 65 °C (30 s) and 72 °C (1 min) for 20 cycles. The second PCR was performed with either CB2 or CG2 and P20EA primers using the same PCR conditions. Amplicons were obtained by amplification of the products from the second PCR using P22EA-ST1 and either CB-ST1-R or CG-ST1-R. The primer sequences are shown in Supplementary Table 16. Following PCR amplification, index (barcode) sequences were added by amplification with Nextera XT index kit v2 setA (Illumina Inc., San Diego, CA, USA). Equimolar concentrations of the indexed amplicon products were mixed and quantified by a Qubit 2.0 Fluorometer (Thermo Fisher Scientific, Waltham, MA, USA). Sequencing was performed using the Illumina MiSeq paired-end platform (2 × 300 bp). The human PBMC total RNA (Takara Bio USA, San Jose, CA, USA) used in this study was derived from normal human peripheral leucocytes pooled from 426 male/female Asians aged 18–54 years old.

**Data analyses**. All paired-end reads were classified by index sequences. Sequence assignment was conducted by determining the sequences with the highest identity in a dataset of reference sequences from the international ImMunoGeneTics information system (IMGT) database (http://www.imgt.org). Data processing, assignment and merging were executed automatically by using a repertoire analysis software program originally developed by DNA Chip Research Inc. (Tokyo, Japan).

Annotated sequence reads were defined as distinct sequence reads within the population of merged sequence reads that had been identified as a BCR gene. The copy numbers of identical annotated reads in each sample were automatically counted using the RG software program and then ranked numerically. In accordance with IMGT nomenclature, the CDR3 nucleotide sequences from a conserved cysteine at position 104 (Cys104) to a conserved phenylalanine at position 118 (Phe118) and the following glycine (Gly119) were translated to presumed amino acid sequences.

**Circos analysis**. Circos software was selected in this study for its high data-to-ink ratio and for its ability to clearly display relational data. Circos open-source software was acquired from www.circos.ca. The V(D)J region recombination data were reformatted using the R statistical programming language to comply with Circos data file requirements. Library sizes were normalized with Circos ideogram (circumference segments) scaling and sizing, permitting comparison of individual subgroups within libraries as well as across disparate libraries. Links, drawn from a V region to its observed J region recombinant partner, were utilized to show the frequency of recombination, with thicker links indicative of higher frequencies of recombination. The ideogram space allotted to the V region subgroup corresponds to the frequency of its observation relative to other subgroups. Analysis of V(D)J recombination was performed with an additional stacked histogram track on each Circos diagram. This track illustrates the relative proportion of each V(D)J recombination as a fraction of the total number of D region sequences observed.

**Detection of somatic hypermutations in human VH and VK regions**. The definition of the clone lineage referred to a set of B cells that were related by descent, which arose from the same V(D)J rearrangement event. The definition of the clonotype refers to a single Ab sequence (CDR1, 2 and 3-joined unique sequence)[40]. NGS reads identical to the CDR1-2-3 sequence were grouped into a single clonotype. Mutations were detected by comparison with germline sequence at every nucleotide position (around 315 nucleotides) and were calculated as a percentage. For example, when 10 reads were recorded in the same clone lineage and three reads had a point mutation at the same nucleotide position, the mutation rate at that position was described as 30%. The mutation rate at every position in the same clone lineage was integrated as a clone linage mutation rate. In addition, different lengths of Ab variable regions in annotated reads existed; therefore, an index of variable region length was set that was the average variable region length in a clone lineage and converted to 100. In this way, clone lineages could be compared at the same magnitude and the index was similar to the amino acid position in the variable region.

**Diversity index**. To estimate BCR diversity in deep sequence data, the Shannon-Weaver index (H′) was calculated using the following Eq. (1):

$$H' = -\sum_{i=1}^{S} \frac{n_i}{N} ln \frac{n_i}{N} \tag{1}$$

where N is the total number of sequence reads, $n_i$ is the number of *i*th annotated reads and S is the species number of annotated reads[41]. A greater H′ value reflects greater sample diversity.

**Phylogenetic analysis of the human Ab repertoire**. Phylogenetic trees (circular dendrograms) were created by alignment of CDRH3 and CDRL3 amino acid sequences using the multiple sequence alignment program and the Neighbour-Joining method[42]. Furthermore, two phylograms of unimmunized and OVA-immunized TC-mAb mice were assembled within one phylogram based on their amino acid sequences. Additionally, copies comprising the same CDRH3 sequences were counted and overlaid on the leaves of circular dendrograms and are shown with a maximum of 50 reads; thereby, as the number of reads increased, the circle in the leaves increased.

**Antigens**. Ovalbumin (OVA) was obtained from Sigma (A7641). To obtain the Trx-EpEX recombinant protein, the extracellular domain of human EpCAM (NM_002354) was amplified by PCR using specific primers. The primer sequences are described in Supplementary Table 16. and subcloned into pET32b (Merck Millipore) using *Eco*RV and *Hin*dIII restriction enzymes (resulting in pET32b-EpEX). To obtain the GST-EpEX recombinant fusion protein, pGEX6P1 (GE Healthcare, Chicago, IL, USA) was modified by insertion of the synthesized DNA using *Bgl*II and *Not*I restriction enzymes (resulting in pGEX-MCS-His). The amplified EpEX fragment was also cloned into pGEX-MCS-His. To produce the AMIGO2 extracellular domain, a DNA fragment of the whole AMIGO2 region (NM_001143668) was amplified by PCR using primers and was subcloned into pET32b (Merck Millipore) using *Hin*dIII and *Xho*I restriction enzymes (resulting in pET32b-AMIGO2-EX). AMIGO2-EX was hard to express in *Escherichia coli* gami B pLysS (DE3); therefore, pET32b-AMIGO2-EX was digested with *Eco*RI (an *Eco*RI site is located near the upstream end of the leucine-rich repeats sequence), blunted using Blunting high, and then digested with *Eco*RV (at a site upstream of AMIGO2-EX) to eliminate the leucine-rich repeats sequence. Therefore, this vector consisted of the Ig-like domain of AMIGO2 (named pET32b-AMIGO2-Ig) and produces Trx-AMIGO2-Ig recombinant protein. After transformation of *E. coli* gami B pLysS (DE3) with each vector, the recombinant proteins were expressed by induction with 1.0 mM Isopropyl-β-D-(-)-thiogalactopyranoside (WAKO) in LB medium. Transformation using the empty vector (pET32b) was also carried out to produce the Tag protein for use in hybridoma screening as a negative control. After harvesting cells and sonication, the recombinant proteins were obtained as inclusion-bodies. Following solubilisation with 6 M guanidine hydrochloride (WAKO) in PBS with 0.1 mM glutathione (oxide form) and 1 mM glutathione (redox form), recombinant protein was purified using Ni-NTA columns with elution using 100 mM imidazole containing 6 M guanidine hydrochloride.

After dialysing the eluted fraction against PBS containing 0.4 M arginine, samples were diluted to ~1 mg/ml and stored at −30 °C. The construction of the pGEX-MCS-His vector and expression and purification of GST-AMIGO2-Ig were carried out using the same procedure as for GST-EpEX.

**Immunization.** Protein antigens (1 mg/ml) were prepared in PBS or in PBS containing 0.4 M arginine, and the volume corresponding to the desired amount of protein was increased to an injectable volume with PBS or PBS containing 0.4 M arginine. This volume was then mixed 1:1 (v/v) with either Freund's or Sigma adjuvant (Sigma Adjuvant S6322; Sigma CFA F5881, Sigma) prepared in accordance with the manufacturer's instructions. For viscous adjuvants, the solution was mixed by repeated passage through a syringe until a smooth emulsion was formed (over 30 min on ice). Injections were performed on 6-week-old male and female mice using a 1-ml glass syringe and a 27-gauge needle. Prime and boost injections were given intraperitoneally (*i.p.*) every 2 weeks. Volumes varied depending on the injection route and experimental requirements and were determined according to the relevant JP Home Office animal license for the procedure. Final boosts were delivered without adjuvant intravenously (*i.v.*) via the tail vein.

**Serum concentration of Abs.** The concentrations of human Igs such as hIgM, hIgG, hIgκ, hIgA and hIgE, and mouse Igs such as mIgM, mIgG, mIgκ and mIgλ were assayed using sandwich ELISA. The concentration of hIgM was assayed using a mouse monoclonal anti-human IgM Ab (Bethyl Laboratories, Montgomery, TX, USA) immobilized on 96-well plates, Nunc MaxiSorp (Thermo) and detected with peroxidase-conjugated mouse anti-human IgM Ab (Bethyl Laboratories). Similarly, hIgG, hIgκ, hIgA, hIgE, mIgM, mIgG, mIgκ and mIgλ were assayed using capture and detector Abs listed in Supplementary Table 17. The samples, standard and Ab conjugates were diluted with sample/conjugate buffer (50 mM Tris, 0.14 M NaCl, 1% BSA, 0.05% Tween 20). 3,3′,5,5′-tetramethylbenzidine (TMB) (Nacalai Tesque, Kyoto, Japan) was used as substrate, and absorbance at 450 nm was measured using a spectrophotometer (BioTek instruments, Winooski, VT, USA). The IgG subclasses were determined using an IgG Subclass Human ELISA Kit (Invitrogen) according to the manufacturer's instructions.

**Serum titre determination.** Serum bleeds taken ~3 days after antigen boost were analysed by ELISA. 96-well immunoassay plates (Nunc Maxisorp) were coated with 100 μl/well of antigen at 0.5 μg/ml in PBS containing 0.4 M arginine overnight at 4 °C. Plates were washed three times with PBS-T (0.05% v/v) and blocked with PBS containing 5% skimmed milk (Difco) for 30 min at room temperature. After being washed again as above, 100 μl of serially diluted serum samples in TBS-T was added to wells and incubated for 1 h at room temperature. After incubation, plates were again washed as above and incubated with 100 μl of anti-human IgG (H + L)-HRP conjugate added at 1/50,000 dilution in TBS-T for 30 min at room temperature. Plates were washed again as above and developed using 100 μl *o*-phenylenediamine dihydrochloride and stopped using 25 μl 1 M $H_2SO_4$. Absorbance was read at 492 nm.

**Hybridoma generation.** Immunized mice were euthanized and their spleens and lymph nodes were harvested, homogenized to single-cell suspensions, and fused with myeloma P3X63Ag8.653 cells using an electro-cell-fusion generator (ECFG21) (Nepagene, Chiba, Japan). Fused hybridoma cells were seeded in 96-well plates. After ~14 days of culture, a primary screen of supernatants was performed by an ELISA. Hybridoma clones that produced EpCAM-specific Abs were identified by an ELISA using GST-EpEX following HAT selection. Cells in the positive wells were picked up and passaged in 96-well plates. Each supernatant was again analysed by the ELISA using Tag, Trx-EpEX and GST-EpEX. Hybridoma clones that reacted with Trx-EpEX and GST-EpEX, but not Tag, were established by two or more limited dilutions. In the case of Balb/c mice, hybridoma cells were screened using Tag and Trx-EpEX. Therefore, hybridoma production from TC-mAb and Balb/c mice was not performed under completely identical conditions. To produce anti-AMIGO2 mAbs, Trx-AMIGO2-Ig was used as an immunogen and GST-AMIGO2-Ig was also used to screen AMIGO2-specific mAbs.

**Hybridoma screening.** Hybridoma cells that produced EpCAM- or AMIGO2-specific Abs were identified by an ELISA and immunocytochemical screening. The human colorectal cell line HCT116 was used for immunocytochemical screening of anti-EpCAM mAbs. CHO cells stably transfected with human AMIGO2 were used to screen anti-AMIGO2-specific mAbs. Cultured cells were harvested from a 10-cm dish and resuspended at $2 \times 10^5$ cells/ml. Each well of a 96-well flat-bottomed plate (TPP) was seeded with 100 μl of cell suspension. Cells were incubated for 2 days at 37 °C in a $CO_2$ incubator. The culture medium was then removed by aspiration and 100 μl of supernatant with Abs was added. After incubation for 1 h on ice, the plates were washed twice with 150 μl of the medium and 100 μl of goat anti-Human IgG (H + L) Cross-Adsorbed Secondary Ab (Abcam, Cambridge, UK) diluted at 1:400 in medium was added. Plates were washed twice with 150 μl of the medium and PBS with 1% (v/v) FBS was added. Plates were scanned under a Keyence BZ-X700 microscope.

**Subclass determination.** The subclasses of obtained human mAbs were determined using antigen-specific ELISA using horseradish peroxidase-conjugated secondary Abs specific for human IgG(H + L), IgG1, IgG2, IgG3, IgG4 and IgM and human Igκ and mouse Igλ. Alternatively, subclasses were determined using the Iso-Gold™ Rapid Human Antibody Isotyping Kit (BioAssay Works, Ijamsville, MD, USA) according to the manufacturer's instructions.

**Humanness score of mAbs.** The amino acid sequence of obtained Abs was analysed using the T20 scoring method, which was developed to calculate the humanness of mAb variable region sequences[23]. A Blast search of the variable region was performed against the T20 Cutoff Human Database available at http://abanalyzer.lakepharma.com. The T20 score for an Ab is obtained from the average of the percent identities of the top 20 matched human sequences. To be considered not immunogenic, T20 scores of the FR and CDR sequences must be above 79, and T20 scores for the FR sequences only must be above 86. Scores near or above these values are predicted to be of low immunogenicity.

**Surface plasmon resonance.** Kinetic analysis was performed using a Biacore T200 (GE Healthcare). Each kinetic run was set up using the kinetic wizard template with six non-zero concentrations in series with at least one of the concentrations in duplicate to check the surface performance and a zero concentration. A blank immobilized surface was used as a reference surface, which was prepared as described in the ligand immobilisation step, but without any ligand. All dilutions were prepared in HBS-EP running buffer (GE Healthcare) at room temperature. Regeneration between each cycle was performed using 10 mM glycine (GE Healthcare) at pH 2.5 for 30 s. The sensor chip protein G or CM5 (GE Healthcare) was used to directly capture human Abs of interest. For kinetic analysis of anti-EpCAM Abs, five concentrations of analyte were used (10, 20, 30, 40 and 50 nM). For kinetic analysis of anti-AMIGO2 Abs, five concentrations of analyte were used (6.25, 12.5, 25, 50 and 100 nM). The data were evaluated post-run using the 1:1 kinetic binding model in Biacore T200 evaluation software to generate $ka$, $kd$, and $K_D$[43].

**Cell staining and flow cytometry.** To evaluate the phenotype of TC-mAb mice, we compared them with age-matched WT mice having a similar genetic background. Bone marrow and spleen tissue, and PBMCs were isolated from adult male and female mice (6–20 weeks of age) using aseptic procedures. Single-cell suspensions were prepared from the bone marrow, spleen, lymph nodes, PECs and PBMCs. Samples were stained with Abs (Supplementary Table 8) and analysed using a CytoFLEX S (Beckman Coulter, Brea, CA, USA). All staining reactions were incubated at 4 °C for 30 min using $1 \times 10^6$ cells in 100 μl staining buffer (PBS with 5% FBS:BD Biosciences Brilliant stain buffer; 1:1) (Franklin Lakes, NJ, USA) containing Mouse Seroblock FcR (Bio-Rad Laboratories, Hercules, CA, USA). Cells were stained with fluorescently labelled isotype controls (Supplementary Table 8) used to detect the positive subsets.

**Immunohistochemistry.** Spleens were fixed with phosphate-buffered 4% paraformaldehyde (Nacalai Tesque, Kyoto, Japan) at 4 °C for 2 h, transferred to 20% sucrose in PBS, frozen in OCT compound (Sakura Finetek) and sectioned. Frozen tissue sections on slides were permeabilized with 50 mM Tris-HCL containing 0.1% Triton-X (pH 8.0) at RT for 10 min and then blocked with Blocking One Histo (Nacalai Tesque) at RT for 10 min. Sections were incubated with a 1:100 dilution of biotin-conjugated anti-CD35 mAb (8C12, BD Biosciences, San Jose, CA, USA) and a 1:100 dilution of Alexa Fluor 647-conjugated GL7 (GL7, BioLegend, San Diego, CA. USA) in TBS-T (1 x Tris-buffered saline and 0.1% Tween 20) containing 5% Blocking One Histo at 4 °C overnight. Sections were then incubated with a 1:200 dilution of Alexa Fluor 594-conjugated streptavidin (BioLegend) and 2 μM DAPI (BioLegend) in TBS-T containing 5% Blocking One Histo at 4 °C for 45 min. Coverslips were mounted with ProLong Gold Antifade reagent (Invitrogen) and sections were analysed with a Zeiss LSM700 confocal microscope (Carl Zeiss, Oberkochen, Germany).

**B cell isolation, survival assay and western blotting.** Splenic B cells were purified by negative selection of CD43+ cells using anti-CD43 magnetic beads (Miltenyi Biotec, Bergisch Gladbach, Germany). The purified B cells ($1 \times 10^7$ cells/ml) were stimulated with 10 μg/mL anti-mouse IgM F(ab)′2 (Jackson ImmunoResearch, West Grove, PA, USA) or 10 μg/mL anti-human IgM F(ab)′2 (Jackson ImmunoResearch) and then lysed in lysis buffer [10 mM Tris-HCl, pH 7.4, 150 mM NaCl, 1% Triton X-100 and 0.5 mM EDTA plus protease and phosphatase inhibitor cocktails (Nacalai Tesque)]. Samples were transferred to polyvinylidene difluoride membranes by electrophoresis and analysed by immunoblotting with antibodies against p-Syk (Tyr525/526) (C87C1) and Syk (D3Z1E) (Cell Signaling Technology, Danvers, MA, USA). The purified splenic B cells were cultured with or without 25 ng/ml BAFF (R&D Systems, Minneapolis, MN, USA) for 24, 48, or 72 h. The frequency of live B cells was assessed using TOPRO3 (Invitrogen) exclusion.

**Repertoire analysis of germinal centre B cells.** Splenic cells were harvested on days 14 and 21 after immunization with OVA and germinal centre B cells

(CD19$^+$CD38$^{lo/-}$GL7$^+$) were sorted by a Moflo XDP (Beckman coulter). More than 1.2 million qualified reads were accumulated from each sample and assembled into merged reads. IgBlast-annotated reads were collated into datasets for subsequent analyses. Saturation of clonotype variations was confirmed in the rarefaction curve of each sample. Additionally, the annotated reads showed that >89% of the Igh and Igk transcripts in germinal centre B cells of TC-mAb mice were productive.

To estimate the degree of accumulation of somatic hypermutations, the diversity of the germline sequence of the V-segment was analysed. First, the V-D-J clone lineage was grouped by the combination of V-D-J segment typing. To compare two samples (e.g., immunized [sample 1] vs. unimmunized [sample 2]), we excluded VDJ cone lineages in which either sample had <100 clones (<100 NGS reads) because the resolution of diversity estimation was insufficient for differentiation analysis. For each V-D-J clone lineage, the statistical significance of the mean difference for %diversity of the V-segment from two samples was calculated by Welch's $t$ test. By setting a $P$-value cutoff of 0.01, the V-D-J clone lineage was categorized as follows: group 1, significantly more diverged in sample 1 M; group 2, more diverged in sample 1, but not significant; group 3, significantly less diverged in sample 1; group 4, less diverged in sample 1, but not significant. The odds score was estimated from a $2 \times 2$ table[44]. The odds score was calculated by the formula: (number of group 1/number of group 2)/(number of group 3/number of group 4). A higher odds score represented a trend of differential SHM accumulation for entire V-D-J lineages.

**Statistical analysis**. Statistical analyses were performed using two-tailed unpaired Student's $t$ test. Differences with $P$-values of <0.05 were considered significant. *$P < 0.05$.

**Reporting summary**. Further information on research design is available in the Nature Research Reporting Summary linked to this article.

## Data availability

Source data are provided with this article. The repertoire analysis data generated in this study have been deposited in the DDBJ database under accession code PRJDB12984. SAMD00442671 (TCK011K_S21) and SAMD00442670 (TCK011MG_S1211) are unimmunized, SAMD00442673 (TCK061hK_S34) and SAMD00442672 (TCK061hMG_S32) are 14 days after immunization. SAMD00442675 (TCK062hK_S35) and SAMD00442674 (TCK062hMG_S33) are 21 days after immunization. SAMD00442677 (TCK031K_S36) and SAMD00442676 (TCK031MG_S3435) are after seven boosters, and SAMD00442669 (TCK041K_S26) and SAMD00442668 (TCK041MG_S4014) are a human PBMC pool sample. Files with "MG" or "hGM" in the file name are analysed for IGH repertoire, and files with "K" or "hK" are analysed for IGK repertoire. The flowcytometry data have been deposited in the FlowRepository database (https://flowrepository.org/) under accession code FR-FCM-Z5ZV corresponding to Fig. 1e, f and Supplementary Fig. 7, FR-FCM-Z5Z8 corresponding to Fig. 7a and Supplementary Figs. 15 and 16, FR-FCM-Z5ZW corresponding to the data of Fig. 7b, c and Supplementary Figs. 17–20, FR-FCM-Z5ZP corresponding to Fig. 7d, e and Supplementary Figs. 21–26, FR-FCM-Z5ZQ corresponding to Fig. 8a–c and Supplementary Figs. 27–29, FR-FCM-Z5ZH corresponding to Fig. 8e and Supplementary Fig. 31, FR-FCM-Z5ZS corresponding to Supplementary Figs. 33 and 34, FR-FCM-Z5ZT corresponding to Fig. 9b and Supplementary Figs. 38 and 39, FR-FCM-Z5ZU corresponding to Supplementary Fig. 35. All data are included in the Supplemental Information or available from the authors upon reasonable requests as are unique reagents used in this study. Source data are provided with this paper.

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

## Acknowledgements

We thank Y. Sato (DNA Chip Research Inc.) for assistance with generating Circos plots, frequency analyses of VH and VK, and circular dendrograms. We thank Dr. S. Aizawa at RIKEN for providing TT2F cell line. We thank Y. Sumida, E. Kaneda, K. Yoshida, M. Fukino, A. Ashiba, Dr. K. Nakamura, T. Kurosaki, F. Adachi, Y. Wang and R. Ohnishi at Tottori University and S. Takehara at Trans Chromosomics Inc. for assistance with generating and maintaining TC-mAb mice, and M. Takami, M. Tanaka, K. Hiramatsu, K. Honma, I. Kanazawa and T. Endo at Trans Chromosomics Inc., Y. Nagashima, and M. Morimura at Tottori University and Y. Okabe at Order-made Medical Research, Inc. for assistance with human Ab production and establishing antigen-specific hybridoma cell lines. We also thank Dr. H. Kugoh, Dr. M. Hiratsuka, Dr. T. Ohbayashi, Dr. F. Okada, Dr. M. Osaki and Dr. T. Ohira at Tottori University and Dr. X. Gao at Harbin Medical University for critical discussions. This study was supported in part by JSPS KAKENHI Grant Number JP21K18256 (Y.B.), the Basis for Supporting Innovative Drug Discovery and Life Science Research (BINDS) from the Japan Agency for Medical Research and Development (AMED) under Grant Number JP21am0101124 (Y.K.), the Science and Technology Platform Program for Advanced Biological Medicine from AMED under Grant Number JP21am0401002 (Y.K. and K.T.), the Basic Science and Platform Technology Program for Innovative Biological Medicine from AMED under Grant Number JP18am0301009 (Y.K.), AMED under Grant Number JP21fk0108141 (Y.K.), AMED under Grant Number JP21gm1610006 (Y.K. and K.T.), and JST CREST Grant Number JPMJCR18S4, Japan (Y.K. and K.T.). This research was partly performed at the Tottori Bio Frontier managed by Tottori prefecture. We thank Jeremy Allen, PhD, and Mitchell Arico from Edanz (https://jp.edanz.com/ac) for editing a draft of this manuscript.

## Author contributions

H.S., Y.K. and S.A. planned the study; S.A., K.K. and A.O. performed cell culture and mouse production experiments; K.K., and M.O. performed cytogenetic analyses; H.S., H. Tanaka, K.Y., G.H., K.M. and H. Takayama performed mAb production experiments. H. S., S.A., Y.N., T.M., S.H., Y.Y. and Y.B. performed FCM analyses and analysed B cell development; K.T. contributed to analysis and discussion of the data; H.S., S.A., T.M. and Y.K. analysed the results, and H.S., S.A., T.M. and Y.K. wrote the manuscript with contributions from each author; Y.M., M.O. and K.T. supervised the study.

## Competing interests

M.O. is a CEO, employee, and shareholder of Trans Chromosomics, Inc. S.A., H.Tanaka, K.M. and H.Takayama are employees of Trans Chromosomics, Inc., Y.M. is a CEO, employee of Order-made medical research, Inc., and the other authors declare no conflicts of interest.
