## [Peer Review File · Nature Communications]

REVIEWER COMMENTS

Reviewer #1 (B biology, germinal center response, VDJ rearrangement) (Remarks to the Author):

In this manuscript, Satofuka and colleagues extend previous work generating an improved version of genetically engineered mice carrying human Immunoglobulin (Ig) heavy (H) and kappa light (L) chain loci. The approach described in this manuscripts involves, as self-replicative platform to insert Ig loci, a murine artificial chromosome (MAC) engineered in CHO cells through Cre- and FLP-mediated recombination to carry the entire human IgH and IgK genomic loci.

Through advanced genetic transfer approaches the engineered MAC (IGHK-NAC) was introduced into mouse embryonic stem cells, which ultimately were used to generate first chimeric and subsequently germline-transmitted animals.

The data reported by the authors show:

- 1) Stable expression of a reporter gene placed in IGHK-NAC in a variety of tissues of TC-mAb mice, indicating stability of the mini-chromosome.**
- 2) Antibody VH and VK-gene repertoire analysis (including CDR3 length) assessed by next generation sequencing of IgH/K transcripts amplified from splenic B cells indicated comparable complexity to that of human peripheral blood B cells. Bias for specific VH genes observed in previous human Ig engineered mice was less evident in IGHK-NAC mice.**
- 3) Accumulation of single nucleotide mutations within VH and VK transcripts amplified from bulk unidentified splenic B cells of IGHK-NAC mice, in particular after immunization, as sign of Ig SHM.**
- 4) All human immunoglobulin isotypes were detected in the serum of IGHK-NAC mice, with human IgG titers, which resulted six times lower than their murine counterparts (measured in wild-type ICR mice).**
- 5) Upon immunization with model antigens, IGHK-NAC responded increasing the mutation rate within IgV genes and started producing antigen-specific soluble antibodies. The response to such antigen was weaker than that observed when the same antigen was used to immunize wild-type mice. Moreover, hybridoma studies revealed a surprising (and unexplained) variability in the response of individual mice (n=2) to the same antigen, with one animals raising preferentially IgM antibodies and the other predominantly Ig class-switched immunoglobulins.**
- 6) Hybridoma technology applied to activated B cells from the spleen of IGHK-NAC mice revealed a good efficiency in the recovery of clones and from a limited set of antibodies produced from them (raised against two antigens), affinity of the recovered monoclonal antibodies was for most in the low nanomolar range.**
- 7) B cell development analyses revealed a number of differences between IGHK-NAC and wild-type mice, including the absence of bone marrow immature B cells, an under representation of follicular B cells at the expense of the marginal zone B cell subset. Expression levels of different mature B cell-associated surface markers including CD19, CD21, Cd23, No information on innate B-1 B cells was provided. Germinal center B cells were insufficiently analyzed, whereas plasma blasts appeared overrepresented in response to immunization with Ova as compared to control mice, possibly suggesting a T-cell independent origin (as GCB cells did not differ between control and IGHK-NAC mice). Measurements of antigen-specific IgG memory B cells are questionable given that in control ICR mice over 50% of all the cells in the spleen resulted IgG positive.**

Main comments

This well-written manuscript follows previous efforts to optimize the generation of humanized mice for production of fully human monoclonal antibodies. The establishment of IGHK-NAC certainly represents a progress, as seen by the complexity of the Ig repertoire analysis which revealed strong similarity with that of human peripheral B cells from health volunteers.

The primary scope to establish this mouse line was to improve the efficacy of production of human monoclonal antibodies possibly selected from a close-to-normal human pre-immune repertoire. The data presented by the authors are suggestive of having achieved this goal. However, there are a number of questions that remain open, which are related to the nature of the antibody responses raised in IGHK-NAC mice.

Specifically:

1) Data shown in Figure 3g establish a strong variability with which individual mice respond to the same antigen. Is this result reflecting the frequency of antigen-specific B cells present in the pre-immune repertoire of individual mice? Could this variability reflect a rather general hypo-responsiveness of the B cells to specific antigens as a result of their recognition through the humanized BCR, which could only in some mice (depending on the original affinity of the BCR for the antigen) exceed a putative threshold level of activation?

2) The production of fully human antibodies IGHK-NAC implies the production of immune complexes in response to immunization with antigens of interest. Human antibody/antigen immune complexes consisting of specific IgG isotypes may show weaker binding to mouse Fcγ receptors, thereby impacting with the kinetics of the GC response and possibly with the relative generation of high-affinity plasma cells vs memory B cells. The availability of monoclonal antibodies of different classes isolated from hybridomas by the authors, raised in response to a specific antigen, should allow the author to determine the extent with which immune complexes formed with the different human Ig isotypes bind to Fcγ receptors on monocyte-macrophages, possibly comparing them to their murine counterparts.

3) The germinal center B cell responses described in the manuscript are poorly characterized.

a. Determining dynamics of GC responses in the spleen (i.e. looking at the % of GC B cells over time) in immunized IGHK-NAC mice would help to determine whether the phases respectively of initial expansion, maintenance and final contraction of GC responses are comparable to those of wild-type mice. In this context, it could be useful to analyze the ratio between DZ vs LZ B cells (assessed for example by flow cytometry), at different time points after the immunization. Any possible difference scored in this comparison would imply differences in the signals emanating from the human BCR in GC B cells and/or in the extent to which hIg immune complexes bind to follicular dendritic cells. The importance of expressing human BCRs for antigen-dependent affinity maturation in GCs of IGHK-NAC mice could be addressed analyzing IGV gene repertoires in sorted GC B cells. Analyzing the complexity of clones expressing unique VDJ rearrangements among GCB cells analyzed at different time points after immunization with a model antigen as Ova, will provide hints on the strength of clonal selection occurring in GCs over time.

b. In figure 4g, authors refer to data on the percentage of Ova-specific IgG+ B cells produced in control and IGHK-NAC mice. In control mice there were over 50% of IgG-expressing B cells among all cells in the spleen. This number is largely over-estimated as the frequency of memory B cells against model antigens ranges between 0.01 and 1% of B cells. Authors should check specificity of their anti-IgG reagents.

4) The changes observed in B cell subset distribution between IGHK-NAC mice and controls are quite remarkable and difficult to interpret

a. The lack of immature B cells in these animals, could be consistent with an accelerated maturation of the cells. The acquisition of IgD on the surface of B cells is linked to the transition between immature and mature B cells. No information was provided on the levels of expression and the relative ratio between IgM and IgD expressed on the surface of

follicular B cells in the lymph node. Providing a comparison between the overall levels of surface BCR (e.g. sIgK levels) in control and IGHK-NAC B cells in the spleen and lymph node is also relevant to exclude any possible difference in the strength of the signals emanating from surface human vs mouse BCRs.

b. The preferential increase in marginal zone B cells together with a reduction of follicular B cells is commonly associated to conditions of more or less strong B cell lymphopenia. Indeed, whereas marginal zone B cells are self-replenishing, follicular B cells require the constant output from the bone marrow to sustain their number. To address a possible lymphopenia caused by the replacement of human BCR with their human counterparts, it would be important to show absolute numbers of MZ, B-2 and B-1 B cells at different ages of the animal.

c. Authors refer to marginal zone B cells as reservoir of human memory B cells. This statement is wrong. Instead, the marginal zone area is a preferential site of homing of memory B cells. Therefore, there is no evidence that an increase in the fraction of MZ B cells is indicative of the formation of antigen-specific memory B cells.

d. The responsiveness of mature peripheral B-2 B cells to BCR-mediated signaling, in particular upon antigen recognition, remains undetermined in IGHK-NAC mice. To test this hypothesis, authors may consider to crosslinking human BCRs and measure in a time-resolved fashion activation of proximal BCR effectors. The manuscript lacks also information on the response of IGHK-NAC B cells to the survival factor BlyS/BAFF. This information could help explain the reduced number of follicular B cells seen in IGHK-NAC mice and the delay in the formation of antigen-specific B cells in response to immunization.

e. The presence of a lambda-1 low allele in IGHK-NAC mice provides the opportunity to assess the strength of IgL allelic exclusion. However, this was not determined. How many IghK-expressing B cells, express also mouse Ig-lambda among newly generated CD93+ transitional B cells? Are there murine Ig-lambda expressing B cells carrying non-functional human Ig κ rearrangements?

f. There is no mention in the manuscript about possible explanations for the substantial change in the surface expression levels of different surface markers in B cells of IGHK-NAC mice, when compared to controls. Figure 4b shows for example lower surface levels for CD19, a key component of the BCR complex, whereas surface CD21 and CD23 levels were clearly higher in IGHK-NAC mice.

g. The manuscript lacks information of innate B-1B cells present preferentially in body-cavity serosa. Where the numbers of B-1B cells (both B1-a and B1-b) present in the peritoneum comparable between ICR and IGHK-NAC mice?

h. In figure 4e, IGHK-NAC mice appear to produce more CD138+ plasmablasts/cells in response to immunization with Ova, than ICR controls. It is not clear at which time point after the immunization, the analysis was performed (this is a missing information that is extended to most data on immunized mice). What are the evidences that such PBs/PCs derive from T-dependent rather than extrafollicular T-independent response? In the latter case, antibodies produced by these PBs are likely to bear low affinity for the antigen and therefore poorly interesting for the monoclonal antibody industry.

In summary, the manuscript by Satofuka and colleagues describe a new mouse strain that has greatly improved capacity to express a broad repertoire of human immunoglobulins. The work suffers from limited biological novelty, but represents a clear step forward in the field of human immunoglobulin transgenesis.

In its present version, the manuscript suffers from limited knowledge on the activation and selection properties in IGHK-NAC mice within of B cells responding to vaccination with antigens of interest. The limited knowledge on the biology of the B cells, together with a rather heterogenous response shown by individual IGHK-NAC animals upon immunization with antigens of interest, prevents a realistic estimation of the utility of this strain for the industrial production of human monoclonal antibodies.

Reviewer #2 (B development, VDJ rearrangement) (Remarks to the Author):

This Ms represents a technical tour-de-force analysis of mice with a novel implementation of a human Ig repertoire. Because of the promise of this system and the interest in humanized antibodies, this paper will be of interest and will motivate additional work in the field. For the most part, the data supports the claims that are made

There are some points to clarify, even if it means additional experiments:

- 1) What is the genetic background of the mice and how inbred are they?**
- 2) for Cell staining and flow cytometry. there is Not enough detail, there is no information as to the Data analysis. the protocol does not include Dead cell exclusion or Fc block, which are standard practices**
- 3) for the OVA-binding cell by FACS, how did they establish specificity? What is gating prior is done to what is displayed?**
- 4) for Sup fig 11, Why so few hIgG in TC-ab? looks like 2 populations in hIgG+OVA+**
- 5) for Figure 4g, not stated that from immunized mice**
- 6) "While the majority (73.8%) of hybridoma clones from individual A (four boosters) produced IgM, those from individual B (seven boosters) produced IgG as the major isotype (86.9%). (Supplementary Table 6)." Why seven boosts required???**
- 7) "Surprisingly, the percentage of OVA-specific B cells in the IgG-positive fraction of lymphocytes was very high in TC-mAb mice (72.3%), but low in WT mice (6.1%) (Supplementary Table 10). " but the # of OVA-specific B cells in the IgG-positive is 64-fold less than WT - why ?**
- 8) Suppl 9 FSC, SSC FSC-W, FSC-H in "antibody combination" column. These are not antibodies. Gating should be shown in a supplemental figure**

Reviewer #3 (Antibody response, repertoire) (Remarks to the Author):

The authors present the construction and characterization of a "trans chromosomal" mouse (TC-mAb mouse) carrying a MAC containing entire human IGH and IGK loci in a background of mouse IgH and IgK KO and IgL low. The authors present data showing that the TC-Mab mice are genetically stable, they reasonably recapitulate human V gene usage, class-switch frequencies, produce antibodies with relatively longer CDRH3 and somewhat surprisingly high SHM. Overall this is an important paper. The amount of work in this manuscript is impressive and the data interesting for a general audience. Having said that the manuscript is quite difficult to follow at times and additionally some revisions will be required before publication:

- 1) p 13" In addition, in contrast to mice with human Ig YAC transgenes¹⁷ and humanized VH regions, which exhibit biased use of the top five frequently used V segments (over 60%), use of the top five V segments in unimmunized TC-mAb mice was lower than 40%, as seen in hPBMCs." The authors need to provide data from the literature to support this claim and importantly add statistics. Along these lines what was the source of the hPBMC data?**
- 2) How does CDRH3 length compare with earlier mouse models (YAC or humanized)**

encoding the human IgH locus?

3) The authors point out that D and J gene usage in the TC-mAb mice is similar to that of human PBMCs. How does it compare to humanized VH mice?

4) Have similar defects in B cell development as those reported here been observed in other advanced humanized models? Please discuss

5). SI Fig 11: Why is the frequency of Ag specific B cells so much higher in the TCR-mAB mice relative to the ICR mice? (Supplementary Fig. 11). The explanation given in the text starting with Because unimmunized TC-mAb mice showed low IgG concentration (Supplementary Table 5)...

6) Fig 4g. The titers do not appear to change in a statistically significant manner between immunizations 5 and 7. Are 7 immunizations really required. Statistical comparison of titers for immunization 5-7 would be helpful. -after all the authors to highlight in the text that more immunizations than in ICR mice were required for producing high quality antibodies.

7). How were convergent CDR3 we're defined? Were these from junctions with same V and J? More specifics are needed. Also the term "convergent" is typically use in reference to junctions found in multiple individuals. Is this what the authors mean here? The relevant text seems quite confusing.

REPLY TO REVIEWERS

(Comments from reviewers are shown in black bold and authors' comments are shown in blue.)

Reviewer #1 (B biology, germinal center response, VDJ rearrangement) (Remarks to the Author):

Main comments

This well-written manuscript follows previous efforts to optimize the generation of humanized mice for production of fully human monoclonal antibodies. The establishment of IGHK-NAC certainly represents a progress, as seen by the complexity of the Ig repertoire analysis which revealed strong similarity with that of human peripheral B cells from health volunteers.

The primary scope to establish this mouse line was to improve the efficacy of production of human monoclonal antibodies possibly selected from a close-to-normal human pre-immune repertoire. The data presented by the authors are suggestive of having achieved this goal. However, there are a number of questions that remain open, which are related to the nature of the antibody responses raised in IGHK-NAC mice.

Reply to reviewer #1:

Thank you for reviewing our manuscript and for your helpful suggestions. This revised version includes additional experimental data and analytical results on the immune response and B cell development in TC-mAb mice. We believe that this revised manuscript addresses your questions, and further enhances the understanding of TC-mAb mice.

In this revised version, the figures and tables, including the supplementary information (Fig. 4, Fig. 5, Supplementary Fig. 8–10 12, 13, 14–41, and Supplementary Table 6, 8–14 and 16) have been updated, and we have modified the Results and Discussion sections to accommodate these findings.

Specifically:

Question 1;

Data shown in Figure 3g establish a strong variability with which individual mice respond to the same antigen. Is this result reflecting the frequency of antigen-specific B cells present in the pre-immune repertoire of individual mice? Could this variability reflect a rather general hypo-responsiveness of the B cells to specific antigens as a result of their recognition through the humanized BCR, which could only in some mice (depending on the original affinity of the BCR for the antigen) exceed a putative threshold level of activation? Is this result reflecting the frequency of antigen-specific B cells present in the pre-immune repertoire of individual mice?

Response:

TC-mAb mice are derived from ICR mice and are outbred mice. We recognise that there is variability in the immune response among individual TC-mAb mice compared with inbred mice. Conversely, the immune response between individual A and individual B was considered to be similar for outbred mice because the immune response of both to Trx-EpEX was observed in the first booster step and that of GST-EpEX was also observed in the second booster step (Fig. 3g). Therefore, we believe that the difference in the immune response between A and B was in the percentage of IgM and IgG subclasses in the clones obtained by the number of booster steps. Because a sequencing analysis of the obtained mAbs derived from individual A was not performed, it is not clear whether the IgM mAbs underwent somatic hypermutation and/or clonal expansion to generate their affinity, and if they have the original affinity to the humanized BCR.

However, we agree that there is a putative threshold level of antibody induction by immunization in TC-mAb mice, which was not similar to that in wild-type mice. We think the delayed immune response (Fig. 3e), the requirement of additional booster steps for some antigens, and the difference in timing of increasing germinal centre B cells (Fig. 4g and Supplementary Table 12) support the unique immune response in TC-mAb mice. We have added the ELISA results of anti-sera against anti-AMIGO2-Ig mAbs to explain how this antigen elicits a sufficient immune response in TC-mAb mice (Supplementary Fig. 12). The following sentence was added to the Discussion section to explain the immune response of TC-mAb mice:

(Page 28, Line 16–18)

When a “putative threshold” that elicits immune response activation exists, TC-mAb mice probably have a higher threshold for some antigens compared with WT mice.

Question 2;

The production of fully human antibodies IGHK-NAC implies the production of immune complexes in response to immunization with antigens of interest. Human antibody/antigen immune complexes consisting of specific IgG isotypes may show weaker binding to mouse Fcγ receptors, thereby impacting with the kinetics of the GC response and possibly with the relative generation of high-affinity plasma cells vs memory B cells. The availability of monoclonal antibodies of different classes isolated from hybridomas by the authors, raised in response to a specific antigen, should allow the author to determine the extent with which immune complexes formed with the different human Ig isotypes bind to Fcγ receptors on monocyte-macrophages, possibly comparing them to their murine counterparts.

Response:

- Activation of Fc gamma receptor by immune complexes with the obtained mAbs

Human Fc gamma R11a is an activating receptor that has high affinity against IgG1, IgG3, and IgG4 (Frontiers in Immunology 10:2968DOI:10.3389/fimmu.2019.02968), and its counterpart is thought to be

mouse Fc gamma RIV (Trends in Immunology, June 2015, Vol. 36, No. 6). Therefore, we analysed mouse Fc gamma RIV activation using the obtained anti-AMIGO2 mAbs including mAMI2C001 (mouse origin), hTNK1C017, and hTNK1C099 (human origin). The immune complex of both mouse and human AMIGO2-specific mAbs elicited Fc gamma RIV signalling in dose-dependent manner, although there was a difference in the intensity (Supplementary Fig. 31). We added this result to the “B-cell development and antigen-specific B cells in TC-mAb mice” subsection of the Results section.

The affinities of the obtained mAbs against AMIGO2 were not identical, which possibly affected the activation efficiency of Fc gamma RIV. However, our results demonstrated that the activation of Fc gamma receptor occurred. We think that an analysis of the difference in the degree of activation depending on the combination of Fc receptors and antibody subclasses is beyond the scope of this experiment.

Question 3;

The germinal center B cell responses described in the manuscript are poorly characterized.

a. Determining dynamics of GC responses in the spleen (i.e. looking at the % of GC B cells over time) in immunized IGHK-NAC mice would help to determine whether the phases respectively of initial expansion, maintenance and final contraction of GC responses are comparable to those of wild-type mice. In this context, it could be useful to analyze the ratio between DZ vs LZ B cells (assessed for example by flow cytometry), at different time points after the immunization. Any possible difference scored in this comparison would imply differences in the signals emanating from the human BCR in GC B cells and/or in the extent to which hlg immune complexes bind to follicular dendritic cells. The importance of expressing human BCRs for antigen-dependent affinity maturation in GCs of IGHK-NAC mice could be addressed analyzing IGV gene repertoires in sorted GC B cells. Analyzing the complexity of clones expressing unique VDJ rearrangements among GCB cells analyzed at different time points after immunization with a model antigen as Ova, will provide hints on the strength of clonal selection occurring in GCs over time.

Answer;

- GC B cells and DZ and LZ

We conducted additional analyses of the GC response including DZ and LZ in the spleens of TC-mAb mice (Supplementary Fig. 26, 28, and Supplementary Table 12).

As the reviewers suggested, initial expansion, maintenance, and final contraction of GC B was detected in TC-mAb mice after primary immunization with OVA. The peak percentage of germinal centre (GC) B formation of B cells was increased and appeared to form later in TC-mAb mice than in WT mice (Fig. 4f–h, Supplementary Fig. 26–28 and Supplementary Table 12). The percentage of GC B cells among lymphocytes over time is also included in Supplementary Table 12. We estimate that this delayed GC formation was

related to the delayed immune response of TC-mAb mice (Fig. 3e). The ratios of DZ/LZ at the primary immune response were also different at least 7 days after immunization (7 weeks of age) (Fig. 4h), but the effect of this difference on the GC response over time remains unclear and further analyses are required.

Altogether, there were differences but no important issues in the GC response. There appeared to be no major dysfunctions in the interaction of GC B with BCR signalling and Tfh cells or the binding of human immunoglobulin-antigen complex to follicular dendritic cells.

We modified the Results section as follows:

(Page 24, Line 7–12)

After primary immunization with OVA, the peak percentage of germinal centre (GC) B cell formation in B cells was increased and appeared to form later in TC-mAb than in WT mice (Fig. 4f–h, Supplementary Fig. 26–28, and Supplementary Table 12) and may be due to the delayed immune response of TC-mAb mice. The ratio of dark/light zone (DZ/LZ) in TC-mAb mice was different at least 7 days after immunization (7 weeks of age).

- IGV gene repertoires in sorted GC B cells

We performed an analysis of the repertoire of GC B cells obtained from immunized TC mAb mice (Supplementary Fig. 36–37). The GC B cells were harvested on days 14 and 21 after immunization with OVA (n=4 or 5), and the pooled samples were applied for repertoire analysis. As immunization progressed, an increase in clonotypes with a large number of reads was observed (Supplementary Fig. 35a). Focusing on the diversity of H and kappa chains, the degree of accumulation of somatic hypermutations (SHMs) was estimated by calculating the odds score, which represents the trend of differential SHM accumulation for entire VDJ lineages; accumulation of SHMs was clearly observed in the kappa chain but was not clear in the H chain (Supplementary Fig. 35b). In the kappa chain repertoire, some of the expanded clonotypes observed in the immunized sample were also expanded on days 14 and 21 (Supplementary Fig. 35c). Therefore, we concluded that the Ab repertoire analysis of the sorted GC B cell fraction indicated clonal expansion of some clone lineages in VK. Furthermore, the VH sequencing of the obtained anti-EpCAM mAbs indicated that the IGHV3-13 clone lineage was sequentially mutated to generate high affinity mAbs (Supplementary Fig. 36). We think that these data support that antigen-specific B cells were effectively produced in the spleens of TC-mAb mice.

b. In figure 4g, authors refer to data on the percentage of Ova-specific IgG+ B cells produced in control and IGHK-NAC mice. In control mice there were over 50% of IgG-expressing B cells among all cells in the spleen. This number is largely over-estimated as the frequency of memory B cells against model antigens ranges between 0.01 and 1% of B cells. Authors should check specificity of their anti-IgG reagents.

Answer:

We apologise for our mistake in using the anti-mouse IgG Ab. As you pointed out, the antibody used in this experiment not only recognises mouse IgG but also mouse IgM. Therefore, we immunized WT (ICR) mice in the same manner as before and re-analysed OVA-specific Ab producing cells using an alternative antibody that specifically recognises mouse IgG. The specificity of human and mouse IgG and that of human IgM and mouse IgG was also confirmed, as shown in Supplementary Fig. 41.

The analysis of OVA-specific IgG B cells in both ICR and TC-mAb mice was updated (Fig 5d, Supplementary Fig. 39–40 and Supplementary Table 14. We changed the Results section of the manuscript as follows:

(Page 25, Line 15–Page 26, Line 4)

Thus, we employed a combination of a fluorescent-labelled anti-human IgG and antigen (OVA) to detect the subset populations of the antigen-specific Ab producing B cells in the spleen (Fig. 5c–e, Supplementary Fig. 39–40, and Supplementary Table 14). The percentage of OVA-specific B cells in the IgG-positive fraction of lymphocytes was higher in TC-mAb mice (70.0%) than in WT mice (54.8%) (Fig. 5e). Taken together, the higher ratio of PB/PC cells and antigen-specific B cells in TC-mAb mice have desirable features for the production of antigen-specific therapeutic mAbs.

Question 4;

The changes observed in B cell subset distribution between IGHK-NAC mice and controls are quite remarkable and difficult to interpret

a. The lack of immature B cells in these animals, could be consistent with an accelerated maturation of the cells. The acquisition of IgD on the surface of B cells is linked to the transition between immature and mature B cells. No information was provided on the levels of expression and the relative ratio between IgM and IgD expressed on the surface of follicular B cells in the lymph node. Providing a comparison between the overall levels of surface BCR (e.g. sIgK levels) in control and IGHK-NAC B cells in the spleen and lymph node is also relevant to exclude any possible difference in the strength of the signals emanating from surface human vs mouse BCRs.

Response:

IgM and IgD expression on the surface of follicular B cells in lymph nodes is shown in Supplementary Fig. 18–19 and Supplementary Table 10. IgD expression was detected in Igκ⁺ follicular mature B cells of TC-mAb mice but most of the cells were weakly positive. Because the antibodies used to detect hIgD and mIgD were different, the degree of expression could not be directly compared. However, we considered that the low expression of IgD may not be a serious defect in B-cell differentiation because the majority (80% or more) of Igκ⁺ follicular B cells express IgM⁺IgD⁺, possibly leading to unique B cell maturation in TC-mAb mice.

The Results section was updated as follows:

(Page 23, Line 12–18)

The number of lymphocytes in the spleens of TC-mAb mice was also decreased by half, whereas that in lymph nodes was higher in TC-mAb mice (Supplementary Table 10). The immature B cell percentages was decreased and, surprisingly, the IgD^{hi} B subset was not detected (Supplementary Fig. 16–19). Analysis of transitional subsets revealed that the T1 and T3 populations were decreased while T2 was increased, which was possibly related to IgD expression^{27–29}, and led to unique B cell maturation in TC-mAb mice.

However, the overall levels of surface Igκ in TC-mAb mice and ICR mice in the spleen and lymph nodes was not distinguishable (Supplementary Fig. 17d, 19c, and Supplementary Table 10). As suggested by the reviewer, we can exclude any possible difference in the strength of the signals emanating from surface human vs mouse BCRs.

b. The preferential increase in marginal zone B cells together with a reduction of follicular B cells is commonly associated to conditions of more or less strong B cell lymphopenia. Indeed, whereas marginal zone B cells are self-replenishing, follicular B cells require the constant output from the bone marrow to sustain their number. To address a possible lymphopenia caused by the replacement of human BCR with their human counterparts, it would be important to show absolute numbers of MZ, B-2 and B-1 B cells at different ages of the animal.

Response:

The absolute number of MZ, B-2, and B-1 B cells in TC-mAb mice and WT mice at different ages were additionally analysed and are shown in Fig. 4d, e, Supplementary Fig. 20–21, and Supplementary Table 11. Previously, we analysed follicular (Fo) B cells as a subset of CD19⁺CD93⁻CD21⁺CD23⁺ and marginal zone (MZ) B cells as a subset of CD19⁺CD93⁻CD21^{hi}CD23⁻, but we changed the marker to Fo B cells (B220^{hi}CD19⁺CD93⁻CD21^{lo}CD23^{hi}) and MZ B cells (B220^{hi}CD19⁺CD93⁻CD21^{hi}CD23^{lo}) to improve the detection accuracy of differentiating B cells. Consistent with the previous experiment (Supplementary Table 9 in the previous version), the Fo/MZ ratio at 20 weeks of age was lower in TC-mAb mice than in ICR mice (Supplementary Table 11). Because there was no significant difference ($P > 0.1$) in the Fo/MZ ratio between TC-mAb and ICR mice at each age ($P=0.14$, $P=0.57$, and $P=0.75$ at 6, 8, and 20 weeks of age, respectively) using the two-tailed unpaired Student's t-test, a tendency of “a preferential increase in marginal zone B cells together with a reduction of follicular B cells” was observed at 20 weeks of age but did not show a significant difference.

The absolute numbers of Fo B and MZ B cells in TC-mAb mice ($n=3$) were significantly reduced compared with WT mice ($n=3$) according to the significantly reduced number of immature B cells in the spleens of TC-

mAb mice. Furthermore, the absolute number of B1a cells (CD19⁺B220^{lo}CD43⁺CD5⁺) was comparable in TC-mAb and ICR mice (Fig. 4e). As suggested by the reviewer, the tendency of a high proportion of MZ may be due to self-replenishment, and TC-mAb mice have symptoms of lymphopenia due to reduced lymphocytes. However, the details remain unclear.

According to these results, we modified the Results section as follows:

(Page 23, Line 18–Page 24, Line 3)

The absolute number of follicular and marginal zone B cells was significantly reduced according to the reduced number of immature B cells in the spleens of TC-mAb mice (Fig. 4c–d, Supplementary Fig. 20–21, and Supplementary Table 10-11).

c. Authors refer to marginal zone B cells as reservoir of human memory B cells. This statement is wrong. Instead, the marginal zone area is a preferential site of homing of memory B cells. Therefore, there is no evidence that an increase in the fraction of MZ B cells is indicative of the formation of antigen-specific memory B cells.

Response:

We appreciate these comments. Accordingly, we amended the relevant sentence in the manuscript as follows:

(Page 25, Line 5–6)

~~In human, marginal zone B cells in the spleen is evidence of SHM in Ig genes and are a reservoir of memory B cells²⁶; therefore,~~ **These data support** that antigen-specific B cells were effectively produced in the spleen of TC-mAb mice.

d. The responsiveness of mature peripheral B-2 B cells to BCR-mediated signaling, in particular upon antigen recognition, remains undetermined in IGHK-NAC mice. To test this hypothesis, authors may consider to crosslinking human BCRs and measure in a time-resolved fashion activation of proximal BCr effectors. The manuscript lacks also information on the response of IGHK-NAC B cells to the survival factor BlyS/BAFF. This information could help explain the reduced number of follicular B cells seen in IGHK-NAC mice and the delay in the formation of antigen-specific B cells in response to immunization.

Response:

As suggested by the reviewer, we tested the activation status of Syk, a BCR proximal kinase, in IGHK-NAC B cells stimulated with anti-BCR by western blot analysis. We confirmed the transient phosphorylation of

Syk and these new data were added to Figure Fig. 4i, j and Supplementary Fig. 29–30. Because BAFF stimulation did not affect Syk activation and B cell survival between TC-mAb and ICR mice, it seems that BAFF signalling is not related to lymphopenia of TC-mAb mice.

We inserted the following sentence in the Results section:

(Page 24, Line 12–15)

In addition, *in vitro* BCR stimulation with anti-human IgM Ab elicited Syk B cell activation, and B cell-activating factor belonging to the tumour necrosis factor family (BAFF) treatment preserved the live B cells (Fig. 4i–j, and Supplementary Fig. 29–30).

e. The presence of a lambda-1 low allele in IGHK-NAC mice provides the opportunity to assess the strength of IgL allelic exclusion. However, this was not determined. How many IghK-expressing B cells, express also mouse Ig-lambda among newly generated CD93+ transitional B cells? Are there murine Ig-lambda expressing B cells carrying non-functional human Igk rearrangements?

Answer;

We analysed the progression of allelic exclusion in human Ig kappa and mouse Ig lambda in unimmunized TC-mAb mice and compared it with ICR mice (13 weeks of age, n=3) (Supplementary Fig. 32–33 and Supplementary Table 13). Our findings were added to the Results section as follows:

(Page 24, Line 16–Page 25, Line 1)

Allelic IgL exclusion clearly progressed between mouse Igλ and human Igκ, with a hlgk:mlgλ ratio of 79.9:10.1 in the mature B cells of TC-mAb mice (Supplementary Fig. 32–33 and Supplementary Table 13).

f. There is no mention in the manuscript about possible explanations for the substantial change in the surface expression levels of different surface markers in B cells of IGHK-NAC mice, when compared to controls. Figure 4b shows for example lower surface levels for CD19, a key component of the BCR complex, whereas surface CD21 and CD23 levels were clearly higher in IGHK-NAC mice.

Response:

In the repeat of this experiment, the expression difference in CD19, CD21, and CD23 was the same as the expression levels between ICR and TC-mAb mice (Fig. 4b and Supplementary Fig. 16 and 18). We appreciate these comments.

g. The manuscript lacks information of innate B-1B cells present preferentially in body-cavity serosa. Where the numbers of B-1B cells (both B1-a and B1-b) present in the peritoneum comparable between ICR and IGHK-NAC mice?

Response:

We performed an analysis of peritoneal exudate cells (PECs) in TC-mAb mice (Fig. 4e, Supplementary Fig. 24–25, and Supplementary Table 11). Our findings were added to the Results section as below:

(Page 24, Line 4–6)

The number of splenic B1a B cells and B1a/b and B2 B cells of PECs did not differ significantly, but appeared to produce fewer B2 cells in TC-mAb mice than in WT mice (Fig. 4e, Supplementary Fig. 22–25, and Supplementary Table 11).

h. In figure 4e, IGHK-NAC mice appear to produce more CD138+ plasmablasts/cells in response to immunization with Ova, than ICR controls. It is not clear at which time point after the immunization, the analysis was performed (this is a missing information that is extended to most data on immunized mice). What are the evidences that such PBs/PCs derive from T-dependent rather than extrafollicular T-independent response? In the latter case, antibodies produced by these PBs are likely to bear low affinity for the antigen and therefore poorly interesting for the monoclonal antibody industry.

Response:

- Information of the immunization state

The previous Figure 4 was separated into Figure 4 and Figure 5 in the revised manuscript, and the old Figure 4e was replaced with Figure 5b. The time point of the data collection is indicated in the legend of Fig. 5b as follows:

(Figure 5 legend, Line 5–6)

Each spleen sample was collected after the seventh and final booster steps of immunization (in mice of approximately 21 weeks old).

- T cell-dependent activation

We have no evidence that the CD138⁺TACI⁺ subset in Fig 5b (old Fig. 4e) included antigen-specific PBs/PCs that the cell response was T cell-dependent; however, GC formation and high-affinity IgG production in the spleen was observed after immunization with adjuvant (FCA) and Ova is known to induce T cell-dependent responses. Although there is no direct evidence, it was considered that T cell-dependent responses were elicited and antigen-specific Ab-producing B cells were included in PBs/PCs in Fig 5b. To demonstrate the T

cell-dependent development of PBs/PCs, T cell KO and/or depletion is required, but we believe that this is beyond the scope of this study.

In summary, the manuscript by Satofuka and colleagues describe a new mouse strain that has greatly improved capacity to express a broad repertoire of human immunoglobulins. The work suffers from limited biological novelty, but represents a clear step forward in the field of human immunoglobulin transgenesis.

In its present version, the manuscript suffers from limited knowledge on the activation and selection properties in IGHK-NAC mice within of B cells responding to vaccination with antigens of interest. The limited knowledge on the biology of the B cells, together with a rather heterogenous response shown by individual IGHK-NAC animals upon immunization with antigens of interest, prevents a realistic estimation of the utility of this strain for the industrial production of human monoclonal antibodies.

Reply to reviewer:

In summary, we conducted additional experiments on B cell development in TC-mAb mice and modified Fig. 4, Fig. 5, Supplementary Fig. 8–10 12, 13, 14–41, and Supplementary Table 6, 8–14 and 16.

Reviewer #2 (B development, VDJ rearrangement) (Remarks to the Author):

This Ms represents a technical tour-de-force analysis of mice with a novel implementation of a human Ig repertoire. Because of the promise of this system and the interest in humanized antibodies, this paper will be of interest and will motivate additional work in the field. For the most part, the data supports the claims that are made. There are some points to clarify, even if it means additional experiments:

Reply to reviewer #2:

Thank you for reviewing our manuscript and for your helpful comments. In this revised version, the figure and tables as well as the supplementary information (Fig. 4, Fig. 5, Supplementary Fig. 8–10 12, 13, 14–41, and Supplementary Table 6, 8–14 and 16) have been updated, and the Results and Discussion sections were amended accordingly. The previous Figure 4 was separated into Figure 4 and Figure 5 in the new version of the manuscript. We believe that this modification will be useful for further understanding the activation and selection process in B cells of TC-mAb mice in response to immunization with an antigen of interest.

1) What is the genetic background of the mice and how inbred are they?

Response:

We inserted the genetic background of the TC-mAb mice in Supplementary information as follows:

(Supplementary information Page 10, Line 17–Page 11, Line 15)

TC-mAb mouse generation.

Chimeric mice were derived from an endogenous Ig KO mouse ES cell subline (6TG-9), C57BL/6 (female) × CBA (male) F1 genetic background with mouse Igh and Igk KO, and were crossed with Jcl:ICR mice (CLEA Japan) with an ICR genetic background. F1 littermates were crossed each other or with Jcl:ICR mice. In the subsequent generations, mice in the same generation were crossed with each other to produce and maintain mice with mouse Igh and Igk KO (HKD mice). F2 mice were obtained as described above, and were crossed with Crl:CD1 mice (Charles River Laboratories Japan), which have the ICR/CD-1 genetic background with Igλ low allele(s). In the subsequent generations, mice in same generation were crossed each other or with HKD mice to produce and maintain mice with mouse Igh and Igk KO, and Igλ low alleles (HKLD mice). Chimeric mice derived from 6TG-9 mES carrying IGHK-NAC were crossed with HKD mice, and offspring were further crossed with HKLD mice to generate TC-mAb mice. TC-mAb mice were maintained by crossing TC-mAb and HKLD mice. In this study, HKLD and TC-mAb mice were of more than 10 and six generations, respectively.

2) for Cell staining and flow cytometry. there is Not enough detail, there is no information as to the Data analysis. the protocol does not include Dead cell exclusion or Fc block, which are standard practices

Response:

We filled the gating strategy and added information about the reagents for cell staining to the Materials section and to the Supplementary Figures and Tables.

3) for the OVA-binding cell by FACS, how did they establish specificity? What is gating prior is done to what is displayed?

Response:

We apologise for our mistake in using the anti-mouse IgG Ab. As you pointed out, the antibody used in this experiment not only recognises mouse IgG but also mouse IgM. Therefore, we immunized WT (ICR) mice in the same manner as before and re-analysed OVA-specific Ab producing cells using an alternative antibody that specifically recognises mouse IgG. The specificity of human and mouse IgG and that of human IgM and mouse IgG was also confirmed, as shown in Supplementary Fig. 41.

The analysis of OVA-specific IgG B cells in both ICR and TC-mAb mice was updated (Fig 5d, Supplementary Fig. 39–40 and Supplementary Table 14. We changed the Results section of the manuscript as follows:

(Page 25, Line 15–Page 26, Line 4)

Thus, we employed a combination of a fluorescent-labelled anti-human IgG and antigen (OVA) to detect the subset populations of the antigen-specific Ab producing B cells in the spleen (Fig. 5c–e, Supplementary Fig. 39–40, and Supplementary Table 14). The percentage of OVA-specific B cells in the IgG-positive fraction of lymphocytes was higher in TC-mAb mice (70.0%) than in WT mice (54.8%) (Fig. 5e). Taken together, the higher ratio of PB/PC cells and antigen-specific B cells in TC-mAb mice have desirable features for the production of antigen-specific therapeutic mAbs.

4) for Sup fig 11, Why so few hlgG in TC-ab? looks like 2 populations in hlgG+OVA+

Response:

Two or more populations in hlgG⁺OVA⁺ (Supplementary Figure 39–40) possibly exhibit different B cell populations with different affinities. We think that the hlgG⁺OVA⁺ signal is determined by two factors: Ab affinity and the number of Abs on the surface of B cells. This is because two or more populations were coincidentally observed in some stained samples, but these were thought to be a mixture of different B cells. We believed that populations with high OVA⁺ signals should contain B cells that produce Abs with high affinity against OVA.

5) for Figure 4g, not stated that from immunized mice

Response:

The “immunized” state of the mice was completed in Figure 5d (old Figure 4g) to indicate that OVA-specific B cells were analysed in immunized mice.

6) “While the majority (73.8%) of hybridoma clones from individual A (four boosters) produced IgM, those from individual B (seven boosters) produced IgG as the major isotype (86.9%). (Supplementary Table 6).” Why seven boosts required???

Response:

We think that the delayed immune response (Fig. 3e), the requirement of additional booster steps for some antigens, and the lower number of cells in germinal centre B cells (updated Fig. 4f and g) support the unique immune response in TC-mAb mice. However, other antigens such as AMIGO2-Ig protein and the ELISA results of anti-sera against anti-AMIGO2-Ig mAbs indicate that this antigen elicited a sufficient immune response in TC-mAb mice by only the second booster step (Supplementary Fig. 12). However, we think that the immune response of TC-mAb mice is weaker than that of WT mice, and if there is a “putative threshold” for immune response activation, that of TC-mAb mice may be higher in response to some antigens.

Therefore, we have added the following sentences to the Results and Discussion sections:

- Results section :

(Page 22, Line 1–10)

Whereas in anti-AMIGO2 mAb production, titres of GST-AMIGO2-Ig-specific human Ig were rapidly increased, reaching a plateau after the second booster step, and the fourth booster was sufficient to induce IgG subclass (Supplementary Fig. 12).

Taken together, we achieved highly efficient generation of antigen-specific human Ab producing hybridomas, although two or more booster immunizations were required in TC-mAb mice for optimal responses, which depended on the antigens. These results also suggest that high population subsets of antigen-specific B cells are contained in the spleen of immunized TC-mAb mice.

- Discussion section:

(Page 28, Line 16–18)

When a “putative threshold” that elicits immune response activation exists, TC-mAb mice probably have a higher threshold for some antigens compared with WT mice.

7) “Surprisingly, the percentage of OVA-specific B cells in the IgG-positive fraction of lymphocytes was very high in TC-mAb mice (72.3%), but low in WT mice (6.1%) (Supplementary Table 10). “ but the # of OVA-specific B cells in the IgG-positive is 64-fold less than WT - why ?

Response:

As mentioned above, the antibody to detect mouse IgG cross-reacted with mouse IgM, which was misleading. We again immunized WT (ICR) mice in the same manner as before and re-analysed OVA-specific Ab-producing cells using an alternative antibody. The specificity of human and mouse IgG and human IgM and mouse IgG was confirmed, as shown in Supplementary Fig. 41. We apologise for the misunderstanding.

The re-analysed data indicated that the percentage of OVA-specific B cells in the IgG-positive fraction of lymphocytes was higher in TC-mAb mice (70.0%) than in WT mice (54.8%) (Fig.5e). Considering the greater induction of PB/PC cells in TC-mAb mice, the higher ratio of antigen-specific B cells was induced in the spleens of TC-mAb mice.

To assess the difference in mAb productivity between TC-mAb and WT mice, the proportion rather than the absolute number of antigen-specific mAb-producing B cells are important in this field. Because more than enough hybridoma cells ($2-8 \times 10^4$ cells) can be obtained from a fraction of the lymphocytes in spleen (1×10^8 cells) by electrofusion and only some of the spleen cells are used for cell fusion, the efficiency of hybridoma production depends on the proportion of antigen-specific B cells in the spleen. This explanation has been added to the legend of Supplementary Table 6 as follows:

(Supplementary Table 6, Page 24, Line 9–12)

To assess the difference in mAb productivity between TC-mAb and ICR mice, we can focus on the proportion rather than the absolute number of antigen-specific mAb-producing B cells. Because a sufficient number of hybridoma cells ($2-8 \times 10^4$ cells) can be obtained from a fraction of the lymphocytes in spleen (1×10^8 cells) by electrofusion, the efficiency of hybridoma production depends on the proportion of antigen-specific B cells in the spleen.

8) Suppl 9 FSC, SSC FSC-W, FSC-H in “antibody combination” column. These are not antibodies. Gating should be shown in a supplemental figure

Response:

Thank you for highlighting these issues. We changed this column to display the correct information (Supplementary Table 9).

Reviewer #3 (Antibody response, repertoire) (Remarks to the Author):

The authors present the construction and characterization of a "trans chromosomal" mouse (TC-mAb mouse) carrying a MAC containing entire human IGH and IGK loci in a background of mouse IgH and IgK KO and IgL low. The authors present data showing that the TC-Mab mice are genetically stable, they reasonably recapitulate human V gene usage, class-switch frequencies, produce antibodies with relatively longer CDRH3 and somewhat surprisingly high SHM. Overall this is an important paper. The amount of work in this manuscript is impressive and the data interesting for a general audience. Having said that the manuscript is quite difficult to follow at times and additionally some revisions will be required before publication:

Reply to reviewer #3;

Thank you for reviewing our manuscript and for your useful suggestions. In this revised version, the figure and tables including the supplementary information (Fig. 4, Fig. 5, Supplementary Fig. 8–10 12, 13, 14–41, and Supplementary Table 6, 8–14 and 16) have been updated, and we have modified the Results and Discussion sections to accommodate these findings. The previous Figure 4 was separated into Figure 4 and Figure 5 in the revised version of the manuscript. We believe that this modification will be useful in further understanding the activation and selection process in B cells of TC-mAb mice in response to immunization with an antigen of interest.

1) p 13" In addition, in contrast to mice with human Ig YAC transgenes¹⁷ and humanized VH regions, which exhibit biased use of the top five frequently used V segments (over 60%), use of the top five V segments in unimmunized TC-mAb mice was lower than 40%, as seen in hPBMCs." The authors need to provide data from the literature to support this claim and importantly add statistics. Along these lines what was the source of the hPBMC data?

Response:

Statistical analyses of the *IgH* repertoire were reported in a YAC-transgenic mouse [ref. 1] and in a genetically chimeric mouse [ref. 2]. However, in another genetically chimeric mouse [ref. 3], the obtained monoclonal antibody-producing hybridomas were only applied to analyse gene segment usage. Thus, we have discussed the human Ig repertoire according to [ref. 1] and [ref. 2].

We described the data from a YAC-transgenic mouse in Figure 1 of [ref. 1] and a genetically chimeric mouse in Supplementary Figure 4 of [ref.2]; however, these claims were not supported statistically. Therefore, we changed the following sentence according to the clearly indicated data in the references:

Old:

"which exhibit biased use of the top five frequently used V segments (over 60%), use of the top five V segments in unimmunized TC-mAb mice was lower than 40%, as seen in PBMCs."

New:

(Page 12, Line 12–14)

"which exhibited biased use of the top three frequently used V segments (over 40%), use of the top three V segments in unimmunized TC-mAb mice was lower than 26%, as observed in PBMCs.

Of note, the top three V segments of TC-mAb mice (V3–23, V2–5, and V3–33) were not identical to those in hPBMCs (V3–23, V1–18, and V3–7) because of the broad distribution of V segment usage. In detail, the top three V segments in a YAC-transgenic mouse, V3–33 (over 20%), V3–23 (significantly over 10%), and V4–59 (approximately 10%), could be detected in Figure 1 of [ref. 1]. And among those in a genetically chimeric mouse, that of HK mice were V3–21, V2–5, and V6–1 (approximately 15%) in Supplementary Figure 4 of [ref. 2].

[ref.1] Longo, N. S., Rogosch, T., Zemlin, M., Zouali, M. & Lipsky, P. E. Mechanisms That Shape Human Antibody Repertoire Development in Mice Transgenic for Human Ig H and L Chain Loci. *J. Immunol.* 198, 3963–3977 (2017).

[ref.2] Lee, E. C. et al. Complete humanization of the mouse immunoglobulin loci enables efficient therapeutic antibody discovery. *Nat. Biotechnol.* 32, 356–363 (2014).

2) How does CDRH3 length compare with earlier mouse models (YAC or humanized) encoding the human IgH locus?

Response:

The CDRH3 length of different mouse models can be compared with those of healthy human donors when the samples were analysed with the same method. Because the determination of a CDRH3 region is dependent on the analysis algorithm, the average distribution of amino acids or nucleotide lengths cannot be directly compared with those previously reported. In this context, it was highlighted in the YAC-transgenic mouse [ref. 1], as follows; "the CDRH3 was significantly shorter compared with that of humans in both the non-productive and productive repertoires ($p \leq 0.003$) (Fig. 5A)". However, it was reported in the genetically chimeric mouse (HK) that the average amino acid length of CDRH3s was one amino acid shorter than those of humans [section "Primary antibody repertoire" of ref. 2].

Regarding the length of CDR3H, it is considered that the human length is possibly reproducible in the mouse when the full-length VDJ region except the constant region of human *IgH* loci is inserted into the genome in both genetically chimeric (HK) and genomically (TC-mAb) mice.

3) The authors point out that D and J gene usage in the TC-mAb mice is similar to that of human PBMCs. How does it compare to humanized VH mice?

Response:

We inserted supplementary figures (Supplementary Fig. 8, 9) that demonstrated the frequency of DH, DJ, and JK usage with a similar format to the genetically chimeric mice [Supplementary Figure 4. of ref. 1]. In TC-mAb mice, the frequency of the usage of these segments was comparable to that in hPBMCs. In the genetically chimeric mice, IGHD3 and IGHD6 of D region and IGHJ4 and IGHJ6 of J region were also frequently used in the heavy chain.

Taking together question 2) and 3), a long CDRH3 length and similar DH and JH segments to those used in the human repertoire were already reproduced in genetically chimeric mice. We added this sentence to the Discussion section as follows:

(Page 27, Line 8–13)

Comparison with Ig-gene segment use in human/mouse chimeric Ab-producing mice (genetically chimeric mice) revealed that the Ig-gene repertoire was slightly or significantly different from that of hPBMCs^{5,7,8,19} and TC-mAb mice, **although a long CDRH3 length and similar DH and JH segments used in the human repertoire were already reproduced in genetically chimeric mice⁷.**

4) Have similar defects in B cell development as those reported here been observed in other advanced humanized models? Please discuss

Response:

As the reviewer pointed out, we recognised that fully human antibody produced in TC-mAb mice influenced B cell development. We defined the difference between genetically chimeric mice, which exhibit no major defects and differences in B cell differentiation, and TC-mAb mice, and added it to the Discussion section as follows:

Old:

Some unique profiles of B cell development and Ab production were also observed in Tc-mAb mice, consistent with reports showing species incompatibility of Ig-genes between human and mouse^{3,11}.

New:

(Page 28, Line 6–12)

In contrast to genetically chimeric mice, which exhibit no major defects as well as differences in B cell differentiation compared with WT mice, some unique profiles of B cell development and Ab production were also observed in Tc-mAb mice. This is consistent with reports showing species incompatibility of Ig-genes between humans and mice^{3,11}, which may due to the interactions between the human Fc region and mouse receptors such as Fc-gamma receptors and Ig-alpha/Ig-beta heterodimer^{3,11}.

5). SI Fig 11: Why is the frequency of Ag specific B cells so much higher in the TCR-mAB mice relative to the ICR mice? (Supplementary Fig. 11). The explanation given in the text starting with Because unimmunized TC-mAb mice showed low IgG concentration (Supplementary Table 5)...

Response:

We apologise for our mistake in using the anti-mouse IgG Ab. As you pointed out, the antibody used in this experiment not only recognises mouse IgG but also mouse IgM. Therefore, we immunized WT (ICR) mice in the same manner as before and re-analysed OVA-specific Ab-producing cells using an alternative antibody that specifically recognises mouse IgG. The specificity of human and mouse IgG and human IgM and mouse IgG was also confirmed, as shown in Supplementary Fig. 41.

The analysis of OVA-specific IgG B cells in both ICR and TC-mAb mice was updated (Fig 5d, Supplementary Fig. 39–40 and Supplementary Table 14). We changed the Results section of the revised manuscript accordingly:

(Page 25, Line 15–Page 26, Line 4)

Thus, we employed a combination of a fluorescent-labelled anti-human IgG and antigen (OVA) to detect the subset populations of the antigen-specific Ab producing B cells in the spleen (Fig. 5c–e, Supplementary Fig. 39–40, and Supplementary Table 14). The percentage of OVA-specific B cells in the IgG-positive fraction of lymphocytes was higher in TC-mAb mice (70.0%) than in WT mice (54.8%) (Fig. 5e). Taken together, the higher ratio of PB/PC cells and antigen-specific B cells in TC-mAb mice have desirable features for the production of antigen-specific therapeutic mAbs.

6) Fig 4g. The titers do not appear to change in a statistically significant manner between immunizations 5 and 7. Are 7 immunizations really required. Statistical comparison of titers for immunization 5-7 would be helpful. -after all the authors to highlight in the text that more immunizations than in ICR mice were required for producing high quality antibodies.

Response:

The source data (see below) revealed that the titres did not appear to change between the fifth and seventh booster steps in individual B (Figure 3g). We also observed that the fifth booster step was enough to obtain hybridoma cells that produce IgG subclass mAbs against antigens such as anti-AMIGO2 mAb (Supplementary Table 12). To avoid the misunderstanding that seven or more booster steps were always required in TC-mAb mice, we added the mAb production data of AMIGO2 as follows:

- Results section

(Page 22, Line 1–10)

Whereas in anti-AMIGO2 mAb production, titres of GST-AMIGO2-Ig-specific human Ig were rapidly increased, reaching a plateau after the second booster step, and the fourth booster was sufficient to induce IgG subclass (Supplementary Fig. 12).

Taken together, we achieved highly efficient generation of antigen-specific human Ab producing hybridomas, although two or more booster immunizations were required in TC-mAb mice for optimal responses, which depended on the antigens. These results also suggest that high population subsets of antigen-specific B cells are contained in the spleen of immunized TC-mAb mice.

- Discussion section

(Page 28, Line 12–18)

For instance, a lower concentration of IgG (Fig. 3a and Supplementary Table 5) and altered B cell development (Fig. 4a–h and Supplementary Table 9–14) in TC-mAb mice compared with WT mice as well as the requirement of two or three additional booster steps for some antigens to elicit an optimal immune response were observed (Fig. 3e). When a “putative threshold” that elicits immune response activation exists, TC-mAb mice probably have a higher threshold for some antigens compared with WT mice.

[Source data of Figure 3g]

7). How were convergent CDR3 we're defined? Were these from junctions with same V and J? More specifics are needed. Also the term "convergent" is typically use in reference to junctions found in multiple individuals. Is this what the authors mean here? The relevant text seems quite confusing.

Thank you for your comments. Accordingly, we deleted the word “convergent” in the manuscript, and “convergent sequences“ was replaced with “expanded clonotypes” in the figure legend of Supplementary Figure 10 (old Supplementary Figure 8).

REVIEWER COMMENTS

Reviewer #1 (Remarks to the Author):

In the revised manuscript, Satofuka and colleagues have included further experiments aiming at better understanding the impact of hIgH and hIgK gene engineering on B cell development, antigen responsiveness and antibody properties of TC-mAb mice rendered defective for mouse IgH and IgK loci.

The new data provide further compelling evidence for the property of TC-mAb mice to generate a broad human antibody repertoire that resembles to a good extent that of human circulating B cells.

The revised manuscript addresses questions and clarifies most issues related to the main defects in peripheral B cell maturation seen in Tc-mAb mice. Yet, the paper falls short in providing a definite immunological and/or molecular basis for altered B cell homeostasis and the unpredictable antibody response to different antigens by TC-mAb mice.

The text could profit from a more focused description of the data and interpretation of the results.

In summary, the revised manuscript by Satofuka and colleagues presents the reader with original data centered around the generation and characterization of genetically engineered immunoglobulin transgenic mice mice generating a highly diversified repertoire of fully humanized antibodies.

The changes in the dynamics of B cell maturation and B cell antibody responses following the introduction in the mouse genome of very large segments of human immunoglobulin heavy and light chain gene loci in the format of a MAC, highlights the need to better understand the complex interplay between membrane-bound and soluble human antibodies with the mouse host, in order to best exploit the potential of this strain for medical applications.

Reviewer #2 (Remarks to the Author):

This is an impressive analysis of a unique approach to generation of human antibodies in the immunologist's favorite furry test tube. The manuscript is much improved through revision and the inclusion of new data. This study is unique enough to be published even if not quite perfect. My recommendations of things to address before publication are:

- 1. In the prior review, a reviewer asked: The presence of a lambda-1 low allele in IGHK-NAC mice provides the opportunity to assess the strength of IgL allelic exclusion. However, this was not determined. How many IghK-expressing B cells, express also mouse Ig-lambda among newly generated CD93+ transitional B cells? Are there murine Ig-lambda expressing B cells carrying non-functional human Igk rearrangements? - I don't think this was answered fully. And in Suppl Fig 32, it is clear that the CD93+ immature B subset is dominated by lambda**
- 2. It is clear that the Mice are of mixed genetic back-ground. This should be re-iterated in the discussion as a caveat to the interpretation of the data.**
- 3. In Supplemental methods, "germinal centre B cells (CD19+CD36+GL7+)" when**

CD19+CD38lo/-GL7+ was used

- 4. For example in Figure 4, how were B1b cells identified? In Suppl Table 11, they are described as CD19+CD23-CD5-. This is not the standard definition of this subset. can the authors demonstrate that most (perC) CD19+CD23-CD5- are CD19+CD43+CD11b+ ?**
- 5. In the Tables that report flow data, the Surface markers column includes FSC,SSC. These are not surface markers, so the column should be re-names**
- 6. Figure 5 - plasmablast -why are there so many in un-immunized Tc-Ab mice ?**
- 7. The staining for OVA-binding B cells by FACS should be deleted. It doesn't add to the paper and the proper controls and validation are not included**
- 8. The Ig sequences from the TC-Ab mouse are compared to human PBMC, but there is no mention of how the PBMC were obtained. A commercial source ? If human samples were obtained directly from patients or local volunteers- there should be IRB-approval or its equivalent. Also, the demographics of the donors should be reported.**
- 9. The gating choices for the flow data should be corroborated by FMO controls, in particular the CD93 gate and the IgK gate**
- 10. Figure sup 14. - are the cells from spleen or bone marrow ? Both are noted for the sample samples**
- 11. Perhaps for future studies - the doublet exclusion in the flow data is not rigorous enough and is critical for some of the populations analyzed**

Reviewer #3 (Remarks to the Author):

I am happy with the authors edits and response to my comments (reviewer #3) and those of the other reviewers. One minor final comment: Please clearly state the definition of clonotype.

REPLY TO REVIEWERS

(Comments from reviewers are shown in black bold and authors' comments are shown in blue.)

REVIEWER COMMENTS

Reviewer #1 (Remarks to the Author):

In the revised manuscript, Satofuka and colleagues have included further experiments aiming at better understanding the impact of hlgH and hlgK gene engineering on B cell development, antigen responsiveness and antibody properties of TC-mAb mice rendered defective for mouse IgH and IgK loci.

The new data provide further compelling evidence for the property of TC-mAb mice to generate a broad human antibody repertoire that resembles to a good extent that of human circulating B cells.

The revised manuscript addresses questions and clarifies most issues related to the main defects in peripheral B cell maturation seen in Tc-mAb mice. Yet, the paper falls short in providing a definite immunological and/or molecular basis for altered B cell homeostasis and the unpredictable antibody response to different antigens by TC-mAb mice.

The text could profit from a more focused description of the data and interpretation of the results. In summary, the revised manuscript by Satofuka and colleagues presents the reader with original data centered around the generation and characterization of genetically engineered immunoglobulin transgenic mice mice generating a highly diversified repertoire of fully humanized antibodies.

The changes in the dynamics of B cell maturation and B cell antibody responses following the introduction in the mouse genome of very large segments of human immunoglobulin heavy and light chain gene loci in the format of a MAC, highlights the need to better understand the complex interplay between membrane-bound and soluble human antibodies with the mouse host, in order to best exploit the potential of this strain for medical applications.

Reply to Reviewer #1:

Thank you for this comment. We agree that the analysis of altered B-cell homeostasis and immune response in TC-mAb is not completely definitive given their mixed genetic background due to outbred strains. This problem can be resolved by backcrossing with inbred mice such as the B6 strain to generate inbred TC-mAb mouse strains. We believe that we would then be able to accurately analyse the effect on B-cell development and immune response when the constant region of the antibody is replaced from mouse to human.

Reviewer #2 (Remarks to the Author):

This is an impressive analysis of a unique approach to generation of human antibodies in the immunologist's favorite furry test tube. The manuscript is much improved through revision and the inclusion of new data. This study is unique enough to be published even if not quite perfect. My recommendations of things to address before publication are:

1. In the prior review, a reviewer asked: The presence of a lambda-1 low allele in IGHK-NAC mice provides the opportunity to assess the strength of IgL allelic exclusion. However, this was not determined. How many IghK-expressing B cells, express also mouse Ig-lambda among newly generated CD93+ transitional B cells? Are there murine Ig-lambda expressing B cells carrying non-functional human Igk rearrangements? - I don't think this was answered fully. And in Suppl Fig 32, it is clear that the CD93+ immature B subset is dominated by lambda.

Reply to Reviewer #2:

It was reported in the Igk-knockout mice with $Ig\lambda^{low}$ mutation that there are fewer immature, and especially mature, $\lambda 1$ B cells and that the production of $\lambda 1$ plasma cells is strongly inhibited¹. This inhibition was not completely recovered as in wild-type mice but was rescued by the human *IGK* locus carried on the artificial chromosome, IGHK-NAC. In addition, Supplementary Figs. 32–33 show that functional human Igk and mouse $Ig\lambda$ are not simultaneously expressed on the surface of CD93+ B cells, indicating that only one kind of antibody light chain is selectively expressed. However, as you pointed out, it is still unclear whether productive mouse $Ig\lambda$ chain-expressing B cells have a non-productive human Igk chain generated during the rearrangement process.

We reanalysed the human Igk chain repertoire in the spleen of TC-mAb mice, focusing on non-productive reads. The numbers of Igk non-productive reads in unimmunized TC-mAb mice, immunized TC-mAb mice, and hPBMCs were 7,684, 18,533, and 14,064, respectively (numbers of total and productive reads are indicated in Supplementary Table 1). The frequency of usage of V segments is compared among them (Figure (a), next page), and also compared between productive and non-productive reads in unimmunized TC-mAb mice (Figure (b)). The results indicate that even non-productive human Igk genes are generated by rearrangement from the entire *IGK* region, with a similar tendency to productive human Igk. In addition, it was detected that a stop codon(s) is located in the non-productive human Igk sequences of TC-mAb mice (Figure (c)). The frequency of stop codon in the CDR3 region of human Igk is about 40% of the non-productive sequences of TC-mAb mice. These findings (Figure (a)–(c)) are not included in the paper.

As described in the main text, TC-mAb mice were generated from ES cells that carry IGHK-NAC containing one copy of human *IGH* and *IGK*, and one human immunoglobulin gene copy is probably maintained in B cells of TC-mAb mice. Therefore, the detection of non-productive human Igk reflects the expression of productive mouse $Ig\lambda$ in those B cells of spleen. The differentiation

of mature B cells, which was inhibited by mouse Igk knockout with Igλ1 mutation, was rescued by the human Igk, and productive human Igk and Igλ were not detected at the same time. Although not directly detected, we believe that these data strongly suggest that human Igk and mouse Igλ undergo allelic exclusion during B-cell development.

(a) Non-productive

(b) Non-productive vs productive in unimmunized TC-mAb mice

(c) IgBLAST of non-productive human Igk (unimmunized TC-mAb mice)

Case #1

```

<-----FR1-IMGT---<-----CDR1-IM---
E I V M T O S P A T L S V S P G E R A T L S C R A S Q S V S
G A A R F A G T G A T G A C C C A G T C C G A G C C A C C T G T C T G T C T C C A G G G A A G A G C C A C C C T C T C G A G G C C A C T C A G A G T G T T A G C
V 98.6% (280/284) IGV3D-15+01 1 .....T.....A...GG.....G 175
E I V M T Q S P A T L S V S P G E R A T L S C R A S Q S V S
V 98.2% (279/284) IGV3D-15+01 1 .....T.....A...GG.....G 90
V 97.5% (277/284) IGV3D-15+02 1 .....T.....A...GG.....G 90

GT---<-----FR2-IMGT---<-----CDR2-IM---
S N L A W Y Q Q K P G Q A P R L L I Y A A S T R A T G I P A
A G C A C T T A G C T G G T A C C A G G A A C C T G C C A G C C C C C A C C A C T C A T C T A C C T G C A C C A C C A G G C C A C T G G T A T C C C G C C
V 98.6% (280/284) IGV3D-15+01 176 .....G.....A... 265
S N L A W Y Q Q K P G Q A P R L L I Y A A S T R A T G I P A
V 98.2% (279/284) IGV3D-15+01 91 .....G.....A... 180
V 97.5% (277/284) IGV3D-15+02 91 .....G.....A... 180

-----FR3-IMGT-----
R F S G S G S G T E F T L T I S S L Q S E D F A V Y Y C * Q
A G G T T C M G T G C A G T G G C T G G G C A G A C C T C C A C C A C T C A C C A G C C T C C A G A G A T T T G C A G T T A C T G T A C G T A G
V 98.6% (280/284) IGV3D-15+01 266 .....T.....A...GG.....G 355
R F S G S G S G T E F T L T I S S L Q S E D F A V Y Y C * Q
V 98.2% (279/284) IGV3D-15+01 181 .....T.....A...GG.....G 270
V 97.5% (277/284) IGV3D-15+02 181 .....T.....A...GG.....G 270

-----CDR3-IMGT---<-----FR4-IMGT---
Y N N W P Y A F C Q G T K L E I K
F A T A T T A C C G C C A G C C T T T G C C A G G G A A C C A C C A G G A G C A C C A C C
V 98.6% (280/284) IGV3D-15+01 356 .....T.....A...GG.....G 407
Y N N W P
V 98.2% (279/284) IGV3D-15+01 271 .....T.....A...GG.....G 284
V 97.5% (277/284) IGV3D-15+02 271 .....T.....A...GG.....G 284
J 97.4% (37/38) IGV32+01 2 .....A..... 39
J 100.0% (33/33) IGV32+02 6 ..... 38
J 100.0% (32/32) IGV32+03 8 ..... 39

```

Case #2

```

<-----FR1-IMGT---<-----CDR1-IM---
D I Q M T Q S P S S L S A S V D R V F I T G R A R H C I T
G A C A T C C A G A G A C C A G T C C C A C C C C T G C A C T G A T A G A C A G A G A C C A C C A C T G C C G G G G C C T C A T T G C A T T A C C
V 95.4% (271/284) IGV1-27+01 1 .....T.....A...GG.....G 183
D I Q M T Q S P S S L S A S V D R V F I T G R A R H C I T
V 89.1% (253/284) IGV1-16+01 1 .....T.....A...GG.....G 90
V 88.7% (252/284) IGV1-16+02 1 .....T.....A...GG.....G 90

GT---<-----FR2-IMGT---<-----CDR2-IM---
N Y S A W Y Q Q K P G K V P R L L I Y A A S T L Q S G V P S
A A T T A T C A G C C T G T A T C A G C C A A C C A G G A A G T T C T A G C T A C T G A T G A T C A T T C A C C A T T G C A A T C A G G G G C C C A C T C
V 95.4% (271/284) IGV1-27+01 184 .....T.....A... 273
N Y S A W Y Q Q K P G K V P R L L I Y A A S T L Q S G V P S
V 89.1% (253/284) IGV1-16+01 91 .....G.....A... 180
V 88.7% (252/284) IGV1-16+02 91 .....T.....A...GG.....G 180

-----FR3-IMGT-----
R F S G S G S G T D F T L T I S S L Q P E D V A T Y Y C Q K
C G G T T C A G C C A G T G A T C G G G A C A G A T T C A C T C A C C A C C A G C C T G C A G C C T G A G A T G T T G C A C T A T T A C T C A A A A G
V 95.4% (271/284) IGV1-27+01 274 .....T.....A... 363
R F S G S G S G T D F T L T I S S L Q P E D V A T Y Y C Q K
V 89.1% (253/284) IGV1-16+01 181 .....T.....A...GG.....G 270
V 88.7% (252/284) IGV1-16+02 181 A A .....T.....A...GG.....G 270

-----CDR3-IMGT---<-----FR4-IMGT---
* N S A P P T F G P G T E V D I K
F A C A C A G T C C C C A T T C A C T T C G C C C T G G G A C A A G T G G A T A C A A C
V 95.4% (271/284) IGV1-27+01 364 .....T.....A...GG.....G 415
N S A P
V 89.1% (253/284) IGV1-16+01 271 .....T.....A...GG.....G 284
V 88.7% (252/284) IGV1-16+02 271 ..T...EA... 284
J 100.0% (38/38) IGV3-01 1 ..... 38
J 86.1% (31/36) IGV34+01 3 .....G...G...G... 38
J 87.1% (27/31) IGV3-01 8 .....A...G...A... 38

```

Figure. Repertoire analysis of non-productive human Igk.

(a) Frequency usage of V segments in non-productive reads of human Igk. (b) Comparison of V segment usage between non-productive and productive reads in unimmunized TC-mAb mice. (c) Nucleotide and amino acid sequences of non-productive human Igk. The stop codon is indicated by a red box.

2. It is clear that the Mice are of mixed genetic back-ground. This should be re-iterated in the discussion as a caveat to the interpretation of the data.

Response:

We added the following sentence in the Discussion section:

Because the TC-mAb mice produced in this study were established from an outbred strain, the detailed analyses above require the generation of animals backcrossed with an inbred strain. (Page 29, line 15–line 18)

We also added the following sentence in the main text and Supplementary information:

Therefore, the TC-mAb mice generated show outbred strains and have a mixed genetic background derived from the ICR strain. (Page 9, line 17–Page 10, line 2, Supplementary information, Page 12, lines 2–3)

3. In Supplemental methods, “germinal centre B cells (CD19+CD36+GL7+)” when CD19+CD38^{lo/-}GL7+ was used

Response:

Thank you for pointing out this mistake. We have corrected it as follows:

The splenic cells were harvested on days 14 and 21 after immunization with OVA, and germinal centre B cells (CD19+CD38^{lo/-}GL7+) were sorted by Moflo XDP (Beckman Coulter). (Supplementary information, Page 15, line 18–Page 16, line 1)

4. For example in Figure 4, how were B1b cells identified? In Suppl Table 11, they are described as CD19+CD23-CD5-. This is not the standard definition of this subset. can the authors demonstrate that most (perC) CD19+CD23-CD5- are CD19+CD43+CD11b+ ?

Answer:

B1b cells in the peritoneal cavity were detected using CD19+CD23-CD5-, with reference to Fig. 4A in ref. 2; this definition is also used elsewhere^{3,4}. Therefore, we added this reference in the manuscript (Page 24, line 7).

5. In the Tables that report flow data, the Surface markers column includes FSC,SSC. These are not surface markers, so the column should be re-names

Response:

We apologize that we failed to correct this despite Reviewer #2 having pointed this out before. I have corrected it appropriately as ‘Gating conditions’ in Supplementary Tables 9–14.

6. Figure 5 - plasmablast -why are there so many in un-immunized Tc-Ab mice ?

Answer:

Although the underlying mechanism remains unclear, it is assumed that the difference between BCR signals mediated by human and mouse constant regions may be responsible for these phenomena. However, we think that it is a favourable feature for producing antibodies using TC-mAb mice.

7. The staining for OVA-binding B cells by FACS should be deleted. It doesn't add to the paper and the proper controls and validation are not included

Response:

We deleted the results of the OVA-binding B cells including Fig. 5c–e, Supplementary Fig. 39–41, and part of Supplementary Table 14 and Table 15. The related sentences in Results, Discussion, and Method sections were also deleted. In the Discussion section, we changed the following and highlighted it in yellow:

Intriguingly, our detailed analyses showed high titre antigen-specific Abs containing all human subclasses (Fig. 3e,h), and a large number of antigen-specific Ab-producing hybridoma cells with SHM and affinity maturation (Fig. 2o and Supplementary Fig. 10, 35, and 36), suggesting that various antigen-specific human Abs were efficiently produced in TC-mAb mice. (Page 28 line 18–Page 29, line 5)

8. The Ig sequences from the TC-Ab mouse are compared to human PBMC, but there is no mention of how the PBMC were obtained. A commercial source ? If human samples were obtained directly from patients or local volunteers- there should be IRB-approval or its equivalent. Also, the demographics of the donors should be reported.

Response:

The human PBMCs used in this study are from a commercial source, and supplier information is indicated as follows:

The human PBMC total RNA (Takara Bio USA, San Jose, CA, USA) used in this study was derived from normal human peripheral leukocytes pooled from 426 male/female Asians aged 18–54 years old. (Page 35, line 18–Page 36, line 2)

9. The gating choices for the flow data should be corroborated by FMO controls, in particular the CD93 gate and the IgK gate

Response:

We agree that the flow data should be corroborated by fluorescence minus one (FMO) controls and compared with negative controls. In this study, we used fluorescence-labelled isotype match controls instead of FOM. We added information of these fluorescence-labelled isotype control antibodies in Supplementary Table 8. Dot plots stained using isotype controls are shown at the top of each plot in Supplementary Fig. 32.

We added the sentence below in the legend of Supplementary Fig. 32.

Dot plots stained with fluorescently labelled isotype controls (Supplementary Table 8) are shown at the top of each dot plot, such as CD93 and CD19, mIg λ and mIg κ , and mIg λ and hIg κ , and are used to detect positive subsets.

Furthermore, the following sentence is added to the Method section.

Cells were stained with fluorescently labelled isotype controls (Supplementary Table 8) used to detect the positive subsets. (Page 46, lines 3–5)

10. Figure sup 14. - are the cells from spleen or bone marrow ? Both are noted for the sample samples

Response:

This figure included the erroneous description of 'Bone marrow' instead of 'Spleen'. It has been corrected. Thank you for pointing this out.

11. Perhaps for future studies - the doublet exclusion in the flow data is not rigorous enough and is critical for some of the populations analyzed

Response:

Thank you for pointing this out. We will conduct doublet exclusion of flow data in future analyses.

[References]

- 1 Kim, J. Y., Kurtz, B., Huszar, D. & Storb, U. Crossing the SJL λ locus into κ -knockout mice reveals a dysfunction of the λ 1-containing immunoglobulin receptor in B cell differentiation. *EMBO J.* 13, 827–834 (1994).
- 2 Skrzypczynska, K. M., Zhu, J. W. & Weiss, A. Positive Regulation of Lyn Kinase by CD148 Is Required for B Cell Receptor Signaling in B1 but Not B2 B Cells. *Immunity* 45, 1232–1244 (2016).

3. Winslow, M. M., Gallo, E. M., Neilson, J. R. & Crabtree, G. R. The calcineurin phosphatase complex modulates immunogenic B cell responses. *Immunity* 24, 141–152 (2006).
4. Kreslavsky, T. et al. Essential role for the transcription factor Bhlhe41 in regulating the development, self-renewal and BCR repertoire of B-1a cells. *Nat. Immunol.* 18, 442–455 (2017).

Reviewer #3 (Remarks to the Author):

I am happy with the authors edits and response to my comments (reviewer #3) and those of the other reviewers. One minor final comment: Please clearly state the definition of clonotype.

Response:

The definition of 'clonotype' is indicated as follows:

The definition of clonotype refers to a single Ab sequence (CDR 1, 2, and 3-joined unique sequence)²². The NGS reads identical to the CDR1-2-3 sequence were grouped into a single clonotype. (Page 16, lines 13–15)